# Modular subgraphs in large-scale connectomes underpin spontaneous co-fluctuation events in mouse and human brains

Elisabeth Ragone[1], Jacob Tanner[2,3], Youngheun Jo [4], Farnaz Zamani Esfahlani[5], Joshua Faskowitz[4], Maria Pope[3,6], Ludovico Coletta[7], Alessandro Gozzi [8] & Richard Betzel [2,4,6 ✉]

Previous studies have adopted an edge-centric framework to study fine-scale network dynamics in human fMRI. To date, however, no studies have applied this framework to data collected from model organisms. Here, we analyze structural and functional imaging data from lightly anesthetized mice through an edge-centric lens. We find evidence of "bursty" dynamics and events - brief periods of high-amplitude network connectivity. Further, we show that on a per-frame basis events best explain static FC and can be divided into a series of hierarchically-related clusters. The co-fluctuation patterns associated with each cluster centroid link distinct anatomical areas and largely adhere to the boundaries of algorithmically detected functional brain systems. We then investigate the anatomical connectivity undergirding high-amplitude co-fluctuation patterns. We find that events induce modular bipartitions of the anatomical network of inter-areal axonal projections. Finally, we replicate these same findings in a human imaging dataset. In summary, this report recapitulates in a model organism many of the same phenomena observed in previously edge-centric analyses of human imaging data. However, unlike human subjects, the murine nervous system is amenable to invasive experimental perturbations. Thus, this study sets the stage for future investigation into the causal origins of fine-scale brain dynamics and high-amplitude co-fluctuations. Moreover, the cross-species consistency of the reported findings enhances the likelihood of future translation.

[1] Neuroscience Program, Oberlin College, Oberlin, OH 44074, USA. [2] Cognitive Science Program, Indiana University, Bloomington, IN 47401, USA. [3] School of Informatics, Computing, and Engineering, Indiana University, Bloomington, IN 47401, USA. [4] Department of Psychological and Brain Sciences and Cognitive Science Program, Indiana University, Bloomington, IN 47401, USA. [5] Stephenson School of Biomedical Engineering, The University of Oklahoma, Norman, OK 73019, USA. [6] Program in Neuroscience, Indiana University, Bloomington, IN 47401, USA. [7] Fondazione Bruno Kessler, Trento, Italy. [8] Functional Neuroimaging Lab, Istituto Italiano di Tecnologia, Center for Neuroscience and Cognitive Systems, Rovereto, Italy. ✉email: rbetzel@iu.edu

A growing body of literature has shown that coordinated brain activity supports ongoing neural, behavioral, and cognitive processes. These activity patterns are constrained by the brain's underlying structural connectivity (SC), whose network configuration organizes brain activity into cohesive and correlated patterns—i.e., functional connectivity (FC)[1,2].

The correlation structure of neural activity is not static; rather, it fluctuates from moment to moment[3,4]. There exist many techniques for tracking these rapid fluctuations, including sliding window[5] and kernel-based methods[6]. However, these approaches generate estimates of time-varying FC (tvFC) that are temporally imprecise. That is, the estimate of FC at time $t$ depends not only on the state of the brain at that instant but also on its state at nearby time points[7].

Recently, we built upon existing frameworks for estimating changes in functional network structure[8–12] to develop a technique–referred to as edge time series—for tracking instantaneous co-fluctuations between pairs of brain regions—i.e., network edges[13,14]. Applying this framework to human functional imaging recordings at rest, we found evidence of global events—brief periods of high amplitude and brain-wide co-fluctuation. We showed that events express subject-specific information[15,16], can be used to approximate static FC and enhance brain-behavior associations[13,17], and can be partitioned into clusters of repeating patterns[15,18] whose relative frequency may be linked to variation in endogenous hormone fluctuations[19]. We refer to studies that calculate and analyze edge time series or the correlation structure of edge time series (so-called edge FC[20–23]) as edge-centric.

Developing insight into the origins of events is the subject of ongoing work[24–26]. First, several studies have shown that event timing is correlated across individuals during movie-watching[13,27,28], suggesting that naturalistic audiovisual stimuli can initiate cascades of neuropsychological processes that act to support the appearance of events in fMRI time series. Second, other studies have demonstrated that events can occur spontaneously in networked dynamical systems[29]. For instance, Pope et al.[30], found that when the underlying constraint matrix—i.e., SC—exhibits modular structure, events naturally occur and that their topography aligns closely with the boundaries of anatomical modules. In aggregate, these findings suggest that event-like activity has both a cognitive underpinning and can also emerge due to modularity in the underlying system structure.

Despite the fact that many studies have applied this edge-centric framework to human imaging data, to our knowledge, it has never been extended to non-human data. Such an extension could prove particularly useful, as the rich set of (sometimes invasive) perturbations[31,32] that can be applied to non-human subjects could help address open controversies and questions surrounding the origins of events and the importance of time-varying coupling between brain areas. Additionally, few empirical analyses have examined the link between SC and high-amplitude edge-centric co-fluctuations (though see[29]).

Here, we take the initial step in that direction, applying event detection to two datasets: first to fMRI BOLD data acquired from 18 anesthetized mice and subsequently to a large human cohort (Human Connectome Project[33]). In line with previous studies, we find evidence of events, show that events are highly predictive of static FC, and can be grouped into hierarchically related co-fluctuation patterns. As demonstrated in Sporns et al.[34], each cluster centroid results in a bipartition of the brain into two disjoint sets of nodes, one positively co-fluctuating and the other negatively co-fluctuating. Finally, we show that the bipartitions are underpinned by highly modular sub-networks in SC, positing an anatomical basis for opposed co-activity. Further, we replicate all of these findings using human functional imaging data. Collectively, our findings set the stage for future, more targeted, and hypothesis-driven investigations into the anatomical underpinnings of network-level co-fluctuations at the fine-scale—i.e., a temporal resolution equivalent to that of the acquisition frame rate.

## Results

The aims of this paper are twofold. First, we seek to replicate, using mouse imaging data, several key findings that have previously been made using data acquired from human participants. Namely, on a per-frame basis, high-amplitude events better recapitulate static FC (correlation networks) than middle-/low-amplitude frames, and events can be meaningfully partitioned into a small set of recurring states or event clusters. The second aim of this paper is to link events detected in both mouse and human imaging data to anatomical connectivity—interareal axonal projections in the mouse and interregional white-matter fiber tracts in humans. In this section, we report the results of these analyses.

**High-amplitude co-fluctuations recapitulate static FC.** One of the first observations made using edge time series was that with only a small subset of high-amplitude frames—putative events—it was possible to accurately reconstruct static FC[13]. Note that here, we define whole-brain FC as the matrix of all interregional correlations. To date, these types of analyses have been carried out largely using functional imaging data acquired from awake humans; whether such events exist in data acquired from animals has been underexplored. Here, we assess whether a similar effect is evident when we apply edge time series to functional imaging data acquired from anesthetized mice.

Our procedure for testing this hypothesis included a series of post-processing analysis steps. First, we transformed the fMRI BOLD time series from $N = 182$ parcels (Fig. 1a) into an edge time series. This procedure involved standardizing (z-scoring) each time series and, for each of the $N(N-1)/2 = 16,471$ pairs, calculating their framewise product (Fig. 1c). The result is a co-fluctuation or edge time series for every pair of nodes whose elements encode the timing, amplitude, and sign of interregional co-fluctuations (Fig. 1d). Notably, the temporal average of a given edge time series is exactly the bivariate product-moment correlation—i.e., FC (Fig. 1b). Thus, edge time series can be viewed as exact decompositions of static FC into time-varying (framewise) contributions.

To detect events, we analyzed all edge time series collectively (Fig. 1e). At each frame, we calculated the global co-fluctuation amplitude as the root mean square (RMS) across all edge time series (Fig. 1f). This step yielded a single time series whose peaks could be detected easily (using MATLAB's `findpeaks.m` function). The amplitudes of these peaks were compared against a null distribution generated by independently circularly shifting parcel time series, recalculating edge and RMS time series, and aggregating peaks of the null RMS time series across 100 runs. Empirical peaks whose amplitude was significantly greater than that of the null distribution were categorized as events (statistical significance was established at the subject level; critical $p$-value adjusted to maintain a false-discovery rate fixed at $q = 0.05$).

Following event detection, which was performed separately for each animal, we aggregated co-fluctuation patterns across animals, computed their pairwise concordance between all patterns (Fig. 1g), and clustered this matrix using modularity maximization. We then averaged the patterns within each of the clusters to obtain cluster centroids globally but also at the single-animal level. We then calculated the similarity (correlation) of these patterns with static FC. Across all animals, the number of

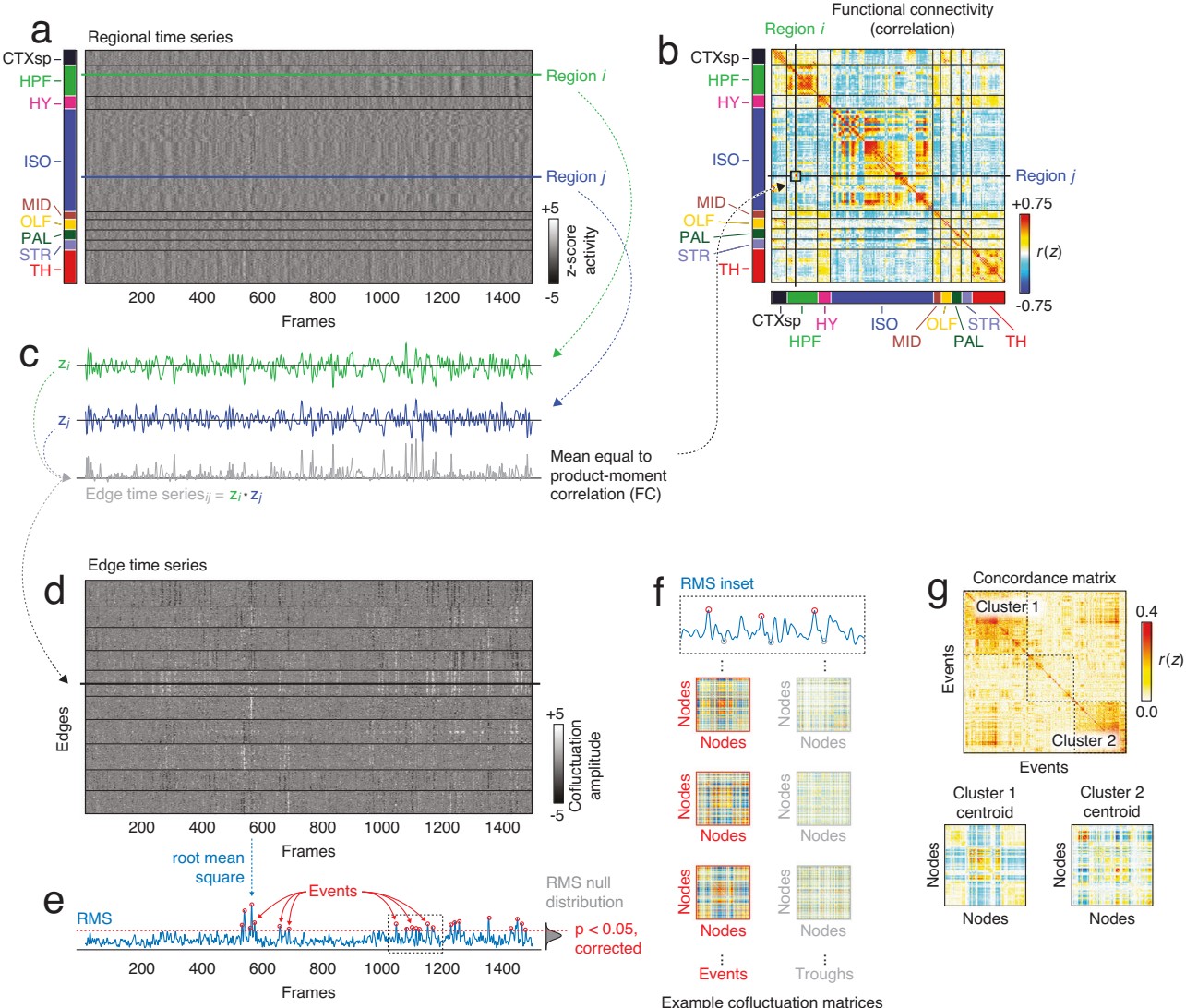

**Fig. 1 Schematic illustrating edge time series construction and clustering. a** Array of parcel time series. Rows and columns correspond to parcels (ordered by anatomical division) and frames, respectively. **b** Whole-brain functional connectivity is typically estimated as the correlation matrix of parcel time series. That is, the weight of the functional connection between nodes $i$ and $j$ is specified as the product-moment correlation coefficient, $r_{ij}$. **c** The procedure for estimating $r_{ij}$ entails z-scoring each parcel time series, calculating their elementwise product—i.e., $z_i(1) \times z_j(1), ..., z_i(T) \times z_j(T)$—and taking the mean of those products (sum divided by the number of samples divided by one). Omitting the averaging step yields the co-fluctuation (or edge) time series $r_{ij}(t) = [z_i(1) \times z_j(1), ..., z_i(T) \times z_j(T)]$, whose elements encode time-varying changes in the weight of the connection between nodes $i$ and $j$. In this panel, we show time series for regions $i$ and $j$ (green and blue curves) and their corresponding edge time series (gray). **d** We can calculate edge time series for all node pairs (edges) in the network and arrange them as rows in an edge-by-frame matrix. **e** Previous studies identified infrequent and short-lived high-amplitude bursts. These bursts or events can be detected by calculating the root mean square (RMS) across all edges at each frame and identifying peaks whose amplitude exceeds that of a null distribution (estimated using the same procedures as empirical RMS but starting with circularly shifted parcel time series). **f** The co-fluctuation patterns expressed during events are very different than those expressed during low-amplitude frames. Here, we highlight approximately 175 frames and show whole-brain co-fluctuation matrices for three local maxima (events; red border) and three local minima (troughs; gray border). **g** Although no two events are identical in terms of co-fluctuation patterns, events can be grouped (broadly) into clusters—i.e., groups of co-fluctuation patterns whose mutual similarity to one another exceeds what would be expected by chance. We detect them by computing the similarity (concordance) between all pairs of events and directly clustering the resulting matrix using a hierarchical algorithm. Here, we highlight two large clusters and their respective centroids (the mean co-fluctuation pattern across all events assigned to each cluster).

events was fewer than the total number of non-event peaks (peaks in the RMS time series that did not reach the statistical criterion for being considered an event), troughs (local minima in the RMS time series), and the total number of frames (by definition). To assess whether events were more similar to these other categories and to control for differences in the number of frames associated with each category, we randomly subsampled a number of frames equal to the number of events from within each category,

averaged the co-fluctuation patterns across these frames, and calculated its similarity with respect to static FC (we performed 100 repetitions of this subsampling procedure; see Fig. 2a for an example of a single set of subsamples and Fig. 2b for the results from 100 sub-samples from one subject).

We then averaged similarity scores across the 100 repetitions for each subject and compared the mean similarity across the four categories of frames: events, non-event peaks, troughs, and

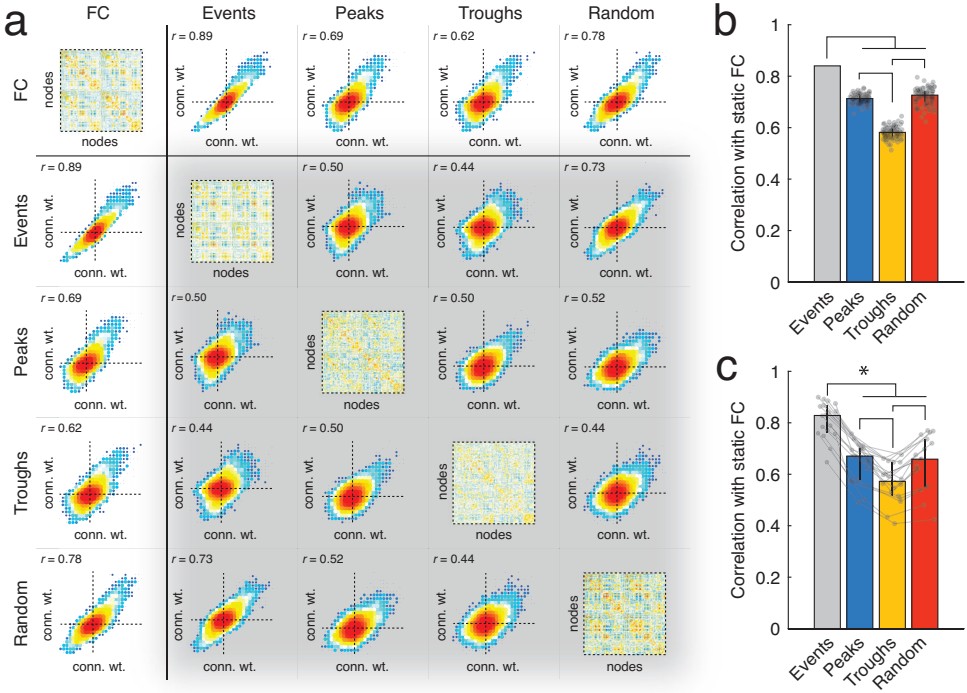

**Fig. 2 Differential correlation between frame categories and static FC.** Previous studies have documented a strong correspondence between static FC and high-amplitude frames (events). Here, we compare four categories of frames: high-amplitude peaks (*Events*), peaks that are not considered high-amplitude (*Peaks*), low-amplitude frames (*Troughs*), and random selections of frames (*Random*). **a** The diagonal shows example co-fluctuation matrices from each of the four categories as well as static FC. The off-diagonal blocks show example scatterplots between each pair of categories. Matrices and scatterplots depicted here come from a single mouse and for the sub-sampled categories, a single sub-sample. **b** Example correlations from a single subject over 100 random sub-samples from within each category. Each sub-sample contained the same number of frames as the number of detected events. **c** Median correlations aggregated across all 18 mice. Lines connect data points from the same mouse. Vertical lines represent ±1 standard deviation. Error bars in panels **b** and **c** correspond to one standard deviation.

random sets of frames (Fig. 2c). We observed that across all subjects, events were significantly more similar to static FC than other frame categories on a per frame basis. The mean similarity of non-event peaks was not significantly different from random samples, while all other frames were significantly more similar to FC than troughs were to FC (false discovery rate fixed at $q = 0.05$ and critical $p$-value adjusted accordingly).

Note that while these observations are consistent with previous findings[13,15], other studies have reported that the highest-amplitude frames may not be optimal in terms of recapitulating static FC (see for example[16,26]). Rather, those studies find that the second-highest amplitude bin outperforms the highest. Why might this be? We investigate this question in Fig. S1. We compared the *binning of all frames* approach of Cutts et al.[16] and Ladwig et al.[26] with *peak binning*, an approach that is more similar to what we report in Fig. 2. We find that using *all frames* approach, we could replicate the effect described by Cutts et al.[16] and Ladwig et al.[26]. With this approach, the extreme bins—the highest- and lowest-amplitude frames—are comprised of co-fluctuation patterns that are temporally proximal to one another. This is likely due to the strong serial correlation of the fMRI BOLD signal, the relative infrequency of events, and the increased number of frames necessary to rise to a high-amplitude event relative to lower-amplitude peaks[18]. In contrast, the *peak binning* procedure exhibited no such bias. Therefore, a possible explanation for the apparent superior performance of the second-highest amplitude bin is that it tends to sample the entire scan session better than the highest-amplitude bin.

Collectively, these findings suggest that a small fraction of high-amplitude frames explain the structure of time-averaged, static FC. Importantly, these findings extend previous lines of research linking events and FC in human brains to mouse models. In the following section, we explore the spatial structure of mouse events in greater detail.

**High-amplitude events can be sub-divided into recurring network states.** Previous studies have shown that high-amplitude and network-level events can be clustered into putative states on the basis of their topographic similarity to one another[15,18,19]. It is unclear whether the same is true for the mouse edge time series data analyzed here. Further, if events can be grouped into clusters, the features that distinguish events in one cluster from those in another are unknown.

To address these questions, we followed the analysis pipeline from Betzel et al.[18]. Briefly, this involved aggregating event co-fluctuation patterns across all subjects, calculating the similarity (Lin's concordance) between all pairs of events (Fig. 3a), and recursively applying modularity maximization (coupled with consensus clustering and a statistical criterion for terminating the recursion) to obtain a hierarchy of statistically significant event clusters (Fig. 3b, c). Note that for two co-fluctuation patterns with identical means and variances, Lin's concordance resolves to the familiar correlation similarity metric. However, unlike correlation, Lin's concordance decreases as the difference between means and variances grows (see Materials and methods).

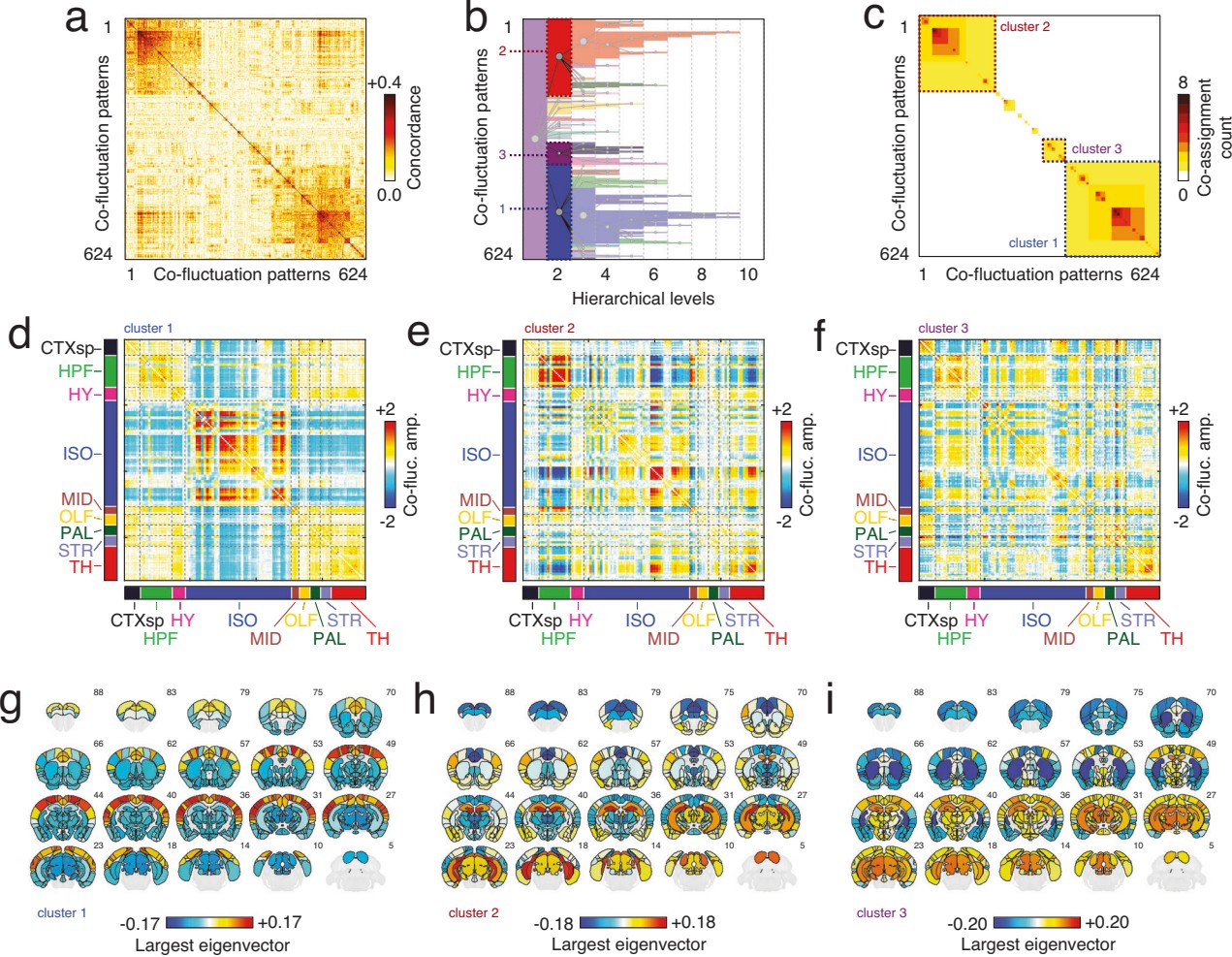

**Fig. 3 Hierarchical clustering of high-amplitude co-fluctuations. a** Concordance matrix. Rows and columns represent events co-fluctuation matrices aggregated across subjects. **b** Hierarchical clustering of co-fluctuations patterns. Gray circles represent clusters—groups of co-fluctuation patterns derived using a variant of modularity maximization—and lines indicate parent–child relationships. **c** Co-assignment matrix (counts) from hierarchical clustering. Panels **d–f** represent centroids for the three large clusters detected at hierarchical level 2. Here, centroids refer to the mean across all co-fluctuation patterns assigned to a given cluster. Panels **g–i** depict the largest eigenvector of each matrix projected back into anatomical space.

Across subjects, we identified 624 putative events ($34.7 \pm 10.2$ events per animal out of 1500 frames; maximum of 49; minimum of 16). The hierarchical clustering procedure grouped the corresponding co-fluctuation patterns into ten hierarchical levels, eight of which were non-trivial (the first and last levels corresponded to a single community and no communities, respectively). For brevity, we focus on the second hierarchical level (the first non-trivial level), which exhibited a total of 12 distinct event clusters, three of which stood out as they collectively accounted for 39.3%, 31.7%, and 9.5% of all co-fluctuation patterns (the next largest cluster contained 5.1% of patterns).

To characterize each cluster further, we estimated and analyzed their respective centroids—i.e., the mean co-fluctuation pattern across all patterns assigned to that cluster. The first event cluster was typified by opposed activation of the isocortex with the midbrain and the hippocampal formation (Fig. 3d, g; see Fig. S2 for details of anatomical labels). This pattern of connectivity has been described at length in mouse imaging literature as a murine analog of the default mode network[35,36]. It also mirrors findings made in the human literature, where the largest event cluster also implicates default mode co-fluctuations[13,15,18,19]. Cluster two was typified by strong co-fluctuations of the hippocampal formation,

an area thought to support memory formation and recall, with components of the isocortex and thalamus, which have been implicated in attention and perception (Fig. 3e, h). Finally, cluster three involved strongly opposed activity of regions in the striatum, isocortex, and the cortical subplate with regions in the hippocampal formation, other isocortical parcels, and midbrain (Fig. 3f, i). Whereas clusters one and two are mostly refined across hierarchical levels, cluster three neatly splits into two subclusters in the third hierarchical level, the first of which emphasized opposed activity of the hippocampal formation with the cortical subplate, striatum, and pallidum, while the second emphasized opposed activity of thalamus with isocortex and to a lesser extent, the cortical subplate (see Fig. S3). Note that the topography of these cluster centroids are largely unaffected by motion correction strategies (Fig. S7).

An alternative and complementary view of high-amplitude events can be obtained by considering their alignment with respect to functional systems obtained using data-driven techniques—e.g., clusters or modules derived from static FC. To this end, we performed a hierarchical decomposition of static FC, revealing multi-level network organization (Fig. S4). Interestingly, the co-fluctuation patterns associated with the event clusters described above neatly align with these static modules.

For instance, the first event cluster corresponds to strong co-fluctuations of module 1 (M1) with the other three modules. The second and third event clusters correspond to opposed co-fluctuations of module 4 (M4) with modules 1 and 2 (M1; M2) and module 2 with, largely, the rest of the brain (Fig. S5). Note that we also recapitulate these findings using high-resolution near-voxel-level data (Fig. S6).

Collectively, and like the results described in the previous sub-section, these findings closely align with analyses of events detected in human functional imaging data. Namely, we show that event co-fluctuation patterns can be described with a relatively small number of clusters, hinting at an approximately finite and discrete repertoire of co-fluctuation states. An important open question, however, is how these patterns emerge from the static underlying anatomical connectivity—i.e., the connectome. We explore this question in the following section.

**High-amplitude co-fluctuation patterns reflect modular sub-divisions of mouse anatomical connectivity.** One of the most important questions in neuroscience is how brain structure relates to brain function. This question can be reframed in the context of this present study: how does the static scaffolding of anatomical connectivity imprint upon measures of brain function–i.e., event co-fluctuation patterns? In Pope et al.[30], the authors detected events in synthetic fMRI BOLD data generated by an anatomically constrained oscillator model. The co-fluctuation patterns associated with these events could be mapped back to the underlying anatomical network, specifically its modular structure. However, the relevance of anatomical modules for high-amplitude edge-level events has never been validated empirically (with human data or otherwise). Moreover, human SC derived from water-diffusion statistics—as used by Pope et al.[30]—exhibits some biases[37–39], including difficulty in accurately tracking interhemispheric fibers[40,41], thereby making a direct empirical replication of Pope et al.[30] less likely.

Instead, we examine the high-amplitude co-fluctuations in mice and their structural underpinnings. Here, the murine connectome was invasively mapped using viral tracers and tract-tracing techniques[42], thereby circumventing some of the known limitations associated with tractography and diffusion MRI data. In addition, the invasive tracing technique allows for the mapping of directed connections, a feature not resolvable using water-diffusion methods.

Our strategy for comparing event co-fluctuations and anato-mical connectivity deviated from that of Pope et al.[30], which depended upon a specific definition of anatomical modules. Instead, our approach was to recover the bipartition of network nodes into positively and negatively fluctuating groups associated with each event cluster centroid[34]. If we were to examine a single co-fluctuation pattern, its bipartition is defined unambiguously. However, for event cluster centroids, which reflect the mean over many co-fluctuation patterns, recovering the bipartition is not as straightforward and requires additional analysis. One possible solution is to apply clustering algorithms—e.g., modularity maximization—to the centroid networks (see Materials and methods). This procedure is not guaranteed to return a perfect bipartition—i.e., exactly two communities—but we can heuristi-cally treat the two largest and anticorrelated modules as estimates of the bipartition. Any other communities, which by definition are smaller, are considered peripheral and not included in subsequent analyses.

Given an estimate of the bipartition, we then imposed this partition onto the network of structural connections and calculated the modularity that it induced[43]. That is, we calculated the extent to which structural connections concentrate within

communities compared to chance. We compared the observed modularity against two null models: one in which we randomly assign nodes to either group, destroying spatial autocorrelations (independent permutation), and another in which we approxi-mately preserve the variogram—i.e., the spatial dependencies—of the original data (geometry preserving permutation)[44].

Interestingly, we found that for the two largest event clusters, their induced modularity exceeded what was expected under both null models (1000 permutation tests; $p < 10^{-3}$ for the indepen-dent permutation model; maximum $p$-value of $p = 0.03$ for the geometry-preserving model; Fig. 4). For cluster three, the modularity was significantly greater than that of the independent permutation model ($p < 10^{-3}$) but not significantly greater than the geometry-preserving model ($p = 0.194$). For event cluster one, these results were not dependent on the resolution parameter used to define the cluster—a free parameter that controls the number and size of detected communities. However, for clusters two and three, there were select ranges where the geometry-preserving model achieved modularity scores consistent with what was observed in the empirical network, underscoring the role of geometry in constraining both the configuration of anatomical connections as well as functional patterns of activity/connectivity (Fig. S8). Interestingly, event co-fluctuation patterns induced greater structural modularity than non-event peaks ($p < 10^{-3}$; Fig. S9), suggesting that the link between structural modularity and co-fluctuation patterns is uniquely strong for events. Indeed, this observation speaks to a more general relationship between co-fluctuation patterns and SC, in which the correspondence between the two is greatest during periods of high-amplitude co-fluctuations (Fig. S10). As before, this effect holds with networks defined at finer spatial scales (Fig. S6).

**Replicating event-module relationships using human imaging data.** One of the aims of this study was to replicate, using mouse imaging data, several key findings that have already been made using human data. In the previous section, however, we identified a link between SC and event co-fluctuations using mouse imaging data. In this section, we reverse our course and seek to replicate this finding in human data. Unlike the mouse connectome, the human connectome is typically reconstructed non-invasively from tractography and diffusion MRI. Although human connectome data has known limitations[37–39], its promise for translation and for understanding uniquely human neu-ropsychiatric disorders is greater. In this section, we replicate the main results from the previous section, demonstrating that event co-fluctuation patterns correspond to modular subgraphs in the human connectome.

Briefly, our replication involved detecting events in resting-state data from the Human Connectome Project. We focused on a subset of the 100 unrelated participants that passed quality checks and motion exclusion criteria (see[14,16]). For each subject and scan (95 subjects × 4 scans each), we performed event detection, aggregating event co-fluctuation patterns across individuals. This procedure resulted in 12854 events, which were subsequently partitioned hierarchically (Fig. 5a). As with the mouse data, we focused on the second hierarchical level, which yielded three large clusters (Fig. 5b, c, e, f, h, i). These cluster centroids were in line with those reported in other studies of event clusters[15,18,19]. Next, using a group-representative SC matrix[45], we calculated the modularity induced by each cluster. As in the previous sections, we compared the observed modularity against a null distribution generated under a permutation-based model in which nodes were randomly assigned to one of the two bipartition communities and a geometry-preserving spin test[46,47]. In all cases, the observed modularity exceeded that of both null models (10,000s

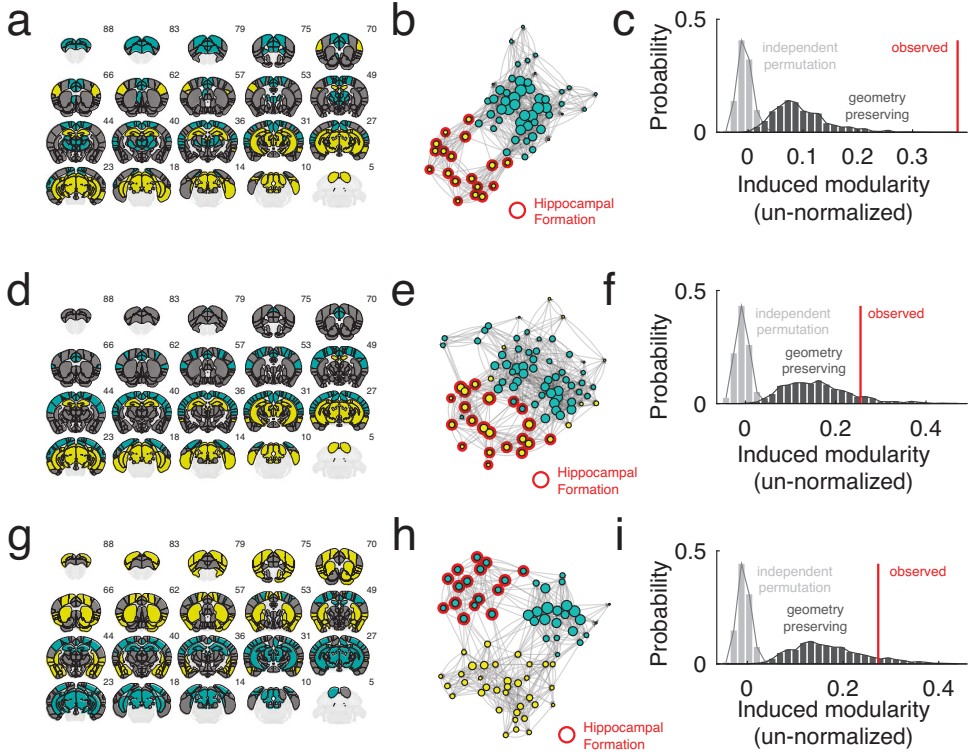

**Fig. 4 Linking high-amplitude events to structural connectivity.** Panels **a**, **d**, and **g** represent bipartition communities for each of the three largest event cluster centroids in hierarchical level 2. Panels **b**, **e**, and **h** force-directed layouts of the induced sub-graph containing only nodes in either of the bipartition communities. The size of nodes is proportional to their weighted degree (strength). Panels **c**, **f**, and **i** show the induced modularity of each sub-graph.

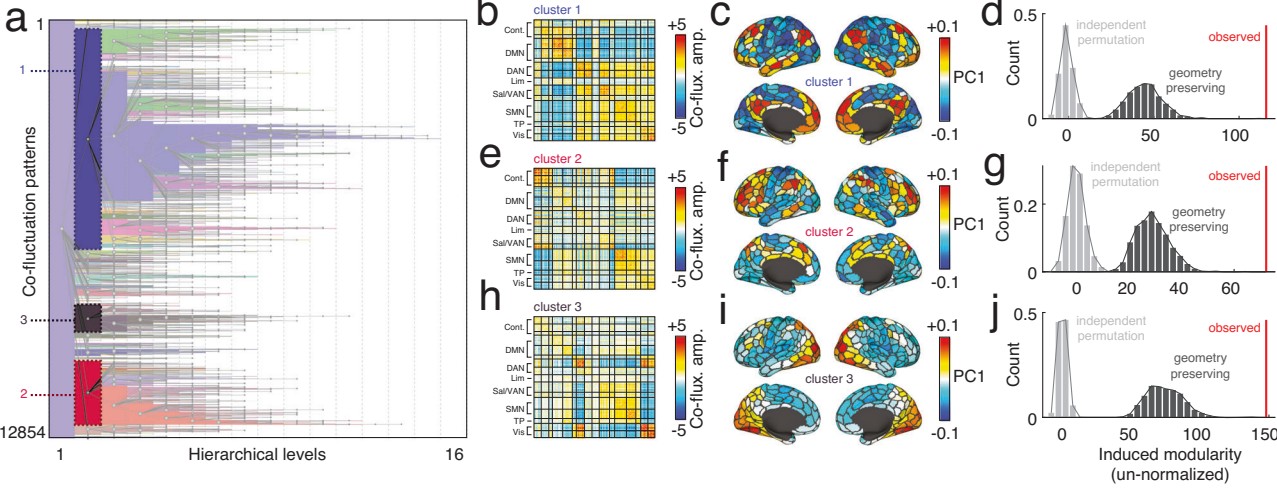

**Fig. 5 Replication of mouse findings using human MRI data. a** Hierarchical clustering of 12854 event co-fluctuation patterns from 95 participants in the Human Connectome Project. Panels **b**, **c**, and **d** show cluster centroids for the three largest event clusters in hierarchical level 2, their projection onto the cortical surface, and the structural modularity induced by a bipartition derived from the co-fluctuation matrix. Here, we include a spatially constrained permutation as a null model—i.e., a spin test[46,47]. Panels **e**–**j** show analogous plots for cluster centroids 2 and 3.

permutations; $p < 10^{-4}$; Fig. 5d, g, j). Note that in the supplement, we also replicate the earlier finding that time-varying structure-function coupling depends on RMS using human imaging data (Fig. S11). Note that here, the claim is not that the co-fluctuation patterns in mice and human brains are analogous in terms of their topography. Rather, the claim is that despite differences in species (mouse *versus* human), level of consciousness (anesthetized versus away), and modality from which the structural

connectomes were derived (diffusion MRI and tractography *versus* axonal tract tracing), the whole-brain co-fluctuation patterns are underpinned by modular sub-networks in the structural connectome.

These results reify, using human imaging data, the observations made using mouse brain networks and suggest that events detected using human fMRI may also be underpinned by modular sub-divisions of the connectome.

## Discussion

Here, we applied edge time series to functional imaging data recorded from anesthetized mice. In agreement with findings made using human imaging data, we find evidence of high-amplitude events. Further, we show that on a per-frame basis, events best predict static FC and can be meaningfully grouped into putative co-fluctuation states. Lastly, we show that the co-fluctuation patterns expressed during events correspond to highly modular anatomical subgraphs, positing a structural scaffold for events to emerge. This study is one of the first to apply edge-centric network methods to non-human imaging data (mouse). Further, this study adds nuance to and contextualizes analogous observations made using human imaging data while paving the way for future work to investigate the origins of high-amplitude BOLD fluctuations using perturbative and possibly invasive experimental techniques.

Numerous studies have shown that patterns of brain activity and connectivity approximately recur within a given scan session[8,48–50]. These observations are not limited to human fMRI but have been made in other mammals, including mice[51–53], and with other imaging modalities, including two-photon and mesoscale calcium imaging[54].

Here, we partition events into non-overlapping clusters using a bespoke hierarchical and recursive variant of modularity maximization. Not only do we find evidence of shared co-fluctuation patterns that recur across time and different mice, but we also find that they can be meaningfully described at different organizational levels. We focus on the coarsest level, where we detect three large and distinct patterns of high-amplitude co-fluctuation that cross-link well-known anatomical and functional divisions of the mouse brain. Critically, the network organization of static FC is well-explained by these high-amplitude states alone, mirroring findings made using human imaging data[15,18,19].

We also examined the activity modes that underlie each of these clusters. Interestingly, they bore a striking resemblance to the results of a recent analysis of high-amplitude mouse activations, in which the authors detected six co-activation patterns (CAPs) that were present in both awake and anesthetized mice[55]. These patterns could be grouped into anti-correlated pairs, such that the spatial patterns of CAPs that correspond to a given pair are approximately anticorrelated with one another.

Notably, the reported CAPs resembled the activity modes that underpinned the event cluster centroids described here (CAPs 3 and 4, 1 and 2, and 5 and 6 mapped onto clusters 1–3, respectively). This is not a coincidence; mathematically, co-fluctuation matrices are calculated as the product of an activation vector with itself transposed. Consequently, an activation pattern (or the same pattern where the sign of each element is flipped) would generate an identical co-fluctuation matrix[34]. Hence, events can be viewed as a connectivity-based analog of the activation-centric CAPs and likely explain the parallels between results presented here and in other studies that analyzed this same dataset[55,56] and others[57–59] using CAPs. In fact, we speculate that in most studies that report CAPs, there will be a two-to-one mapping of CAPs to event co-fluctuation patterns. Both are detected as frames with exceptionally high-amplitude activity (CAPs) or co-fluctuations (edge time series), and the mathematical relationship between activations and edges stipulates that every co-fluctuation pattern can arise from two perfectly anti-correlated activation maps.

In the context of these observations and the long history of detecting and tracking network states, our observations suggest that while time-varying FC is, in principle, a high-dimensional construct, its temporal evolution can be described in terms of relatively few relevant dimensions—i.e., transitions to and from different network states.

We note, however, that the functional/behavioral relevance of these states remains undisclosed. However, given the anesthetized state of the animals, the link to ongoing behavior is tenuous. Rather, they may play a role associated with homeostatic processes[60]. Future studies should investigate this question in greater detail.

Many studies have shown that anatomical connectivity serves as a powerful constraint on both static[1,2,61–63], as well as time-varying FC[64–67]. To date, however, few studies have examined structure-function relationships when the function was defined using edge time series (though see[29,68] for examples).

Here, we study structure–function relationships using an invasively mapped, directed, and weighted connectome through two complementary approaches. First, we show that high-amplitude events result in a division of network nodes into positively and negatively co-fluctuating clusters and that this bipartition is undergirded by highly segregated structural modules. This observation is directly in line with Pope et al.[30] and other studies demonstrating that in modular networks, modules easily synchronize[69–71], possibly building to network-wide events. The present study, therefore, represents the first empirical corroboration of Pope et al.[30]. Importantly, we verified that this effect was not driven by module size or spatial extent (distance), the parameter combinations needed to detect the bipartition, and was replicated it using human imaging data.

Secondly, in a supplementary analysis, we tracked moment-to-moment structure-function coupling as the correlation between instantaneous co-fluctuation matrices and SC. This analysis was similar to Liu et al.[68], who predicted co-fluctuation patterns using stylized measures of inter-regional communication capacity[72]. Here, we opted for a simpler metric of coupling based on edge weight correlations, discovering modest coupling across time. Interestingly, however, we found that structure-function coupling was maximized during high-amplitude frames, supporting our previous finding that event co-fluctuations are well-aligned with anatomical connectivity.

Perplexingly, this observation deviates from previous findings. A number of studies using sliding-window methods for tracking time-varying connectivity reported the presence of hyper-connectivity states when the global coupling is disproportionately strong[73,74]. In these studies, hyper-connectivity corresponded to decoupling from the underlying anatomical network[65,66], with stronger coupling observed in lower amplitude states. There are, of course, a number of possible explanations. For instance, edge time series and sliding windows measure global amplitude in different ways; whereas sliding window estimates of time-varying connectivity define edge weights as correlation coefficients, effectively placing an upper limit on the global mean connectivity, edge time series have no such bound. Therefore, high-amplitude/hyperconnected time points may not align across methods. This agrees with previous studies that reported only a modest correspondence between time-varying networks estimated using those two techniques[7].

Notably, the observation that modular network structure plays a key role in shaping (near) synchronization patterns is a well-documented phenomenon in complex systems and network science[71,75]. Assortative modules are characterized by dense, recurrent connectivity patterns, allowing for the self-excitation of individual modules[76]. On the other hand, the relative segregation of modules from one another ensures that synchronization effects remain localized to subsets of modules, rarely inducing global synchronization stats[69] (though in clustered networks, a path to global synchrony becomes possible through the synchronization of the clusters themselves[77]). Though not explicitly tested here, these mechanisms explain observations made in previous

simulation studies[29,30] and align with the empirical findings reported here.

Our findings suggest that the modular structure of long-distance anatomical connectivity helps shape the organization of high-amplitude co-fluctuation patterns. Our analyses are motivated by a century of neuroscientific observations, demonstrating that perturbations to long-distance connectivity result in acute changes to function[78,79]. However, a recent study has challenged this very premise, positing that long-distance connections play only a small role in shaping brain function[80]. Rather, that study claims that brain function emerges from modes derived from short-range, regular connectivity patterns that reflect the brain's intrinsic geometry. While this perspective has yet to be fully embraced by the neuroscientific community[81,82], high-amplitude events could serve as another feature along which to compare and adjudicate between the two theories. That is, given the apparent ubiquity of events in large-scale imaging data, it would be interesting to assess whether events appear in simulated time series generated by the geometric models and, if so, to explore their structural underpinnings. We leave this exploration for future studies.

Here, we show that events in both mouse and human functional imaging data are undergirded by modular structural networks. This cross-species convergence is particularly interesting given differing levels of consciousness—mice were anesthetized while human participants were awake. Although previous studies have suggested that event timing is modulated by sensory inputs[13,27,28], this observation indicates that consciousness is not a requirement for the emergence of events. Broadly, this observation is aligned with the work of Pope et al.[30], who demonstrated that events can occur in the absence of exogenous input if the structural network is sufficiently modular.

On the other hand, these results raise the question: exactly what is the purpose of events? Clearly, they have both online and offline contributions. Focusing on the offline component, one possibility is that spontaneous co-fluctuations (including events) reflect Hebbian tuning of synaptic weights, reinforcing network modules[60]. Another possibility, also outlined by Laumann and Snyder[60], is that spontaneous co-fluctuations have a restorative, homeostatic effect and return neuronal populations towards excitatory/inhibitory balance. Notably, these processes occur offline and could, therefore, also explain the observation of events in the brains of anesthetized mice.

More broadly, these observations underscore the need to more clearly elucidate not only the network-level mechanisms that support the emergence of events, but the contributions of neurobiological, physiological, molecular, and environmental factors, as well. Addressing this question requires challenging cross-disciplinary and multi-scale research but could shed light on the role(s) of events in normative brain function.

This study presents a number of opportunities for future work but also suffers from some limitations. Most notably, to the best of our knowledge, this study represents the first to examine event structure in a model organism. This extension of the edge time series is critical; its application to human subjects has left a number of questions unanswered. Namely, it remains unclear why events occur and what brain/physiological processes they support. These questions are difficult to answer in the absence of direct, and possibly invasive, measurements and perturbations.

Additionally, future work should consider extending the edge-centric approach from fMRI to other imaging modalities—e.g., widefield calcium imaging[83] or voltage-sensitive indicators[84]. Although the fMRI BOLD signal enjoys a broad agreement with these signals[85], its neuronal provenance is oftentimes unclear and indirect. Relatedly, future studies should also investigate edge time series and events in recordings made in non-mammal brains

—e.g., larval zebrafish[86,87]—where whole-brain activity can be recorded at single-cell resolution[88] and for which anatomical connectivity is mapped at an areal level[89].

There are also a number of limitations associated with this work. For instance, due to the inability to map axonal projections at the level of individual mice (the connectome from Oh et al.[42] is a composite of many brains), all structure–function associations were calculated with respect to a single reference connectome. There are many challenges and issues associated with group or consensus-based connectomes in human imaging[45] and even if such issues are successfully mitigated with invasive tract tracing, the current analysis makes it impossible to calculate measures of structure-function correspondence for individual brains.

Our study relies on connectome data reconstructed using two distinct techniques: invasive tract tracing in mice and non-invasive tractography and diffusion MRI in humans. Although we report converging evidence across both modalities showing that high-amplitude events are underpinned by modular structural networks, there are a number of dissimilarities worth noting explicitly. Critically, the mouse connectome is directed, i.e., $W_{ij} \neq W_{ji}$, and capable of resolving asymmetries in connection weight, whereas the human connectome is not. Additionally, it is well-established that tractography algorithms struggle to resolve crossing fibers[90], recover superficial tracts[38], and, even across well-established pipelines, can lead to variability in tract reconstructions[39]. Nonetheless, the non-invasive nature of MRI means that human connectomes can be reconstructed at a whole-brain level for individual participants. In summary, we identify comparable effects using both techniques but note that in future studies, it may be advantageous to focus on dissimilarities—e.g., the specific contributions of directed connections.

Yet another potential limitation concerns the null models used to assess the statistical significance of induced modularity. For the mouse data, we compared the observed modularity against an ensemble of equal-sized subgraphs with equal-sized communities but otherwise selected at random. This model does not preserve the geometry of the observed bipartition—i.e., the randomly generated communities are much less spatially compact with no guarantees of spatial contiguity. This is an important feature in brain networks, as both anatomical and functional connection weights are distance dependent[91–96]. Addressing this limitation with mouse data is not straightforward. The accepted approach uses spin models to project spatial maps to a spherical surface, rotate the surface randomly, and then project the rotated values back to anatomy[47]. Here, we work with mouse volumetric data, making the implementation of spin tests challenging. To partially address this concern, we replicate our findings using surface-based human imaging data where spin tests are easily performed. There, we found that, like the unconstrained permutation test, the observed modularity exceeded that of the null distribution, suggesting that the mouse results may generalize as well. However, future work is needed to confirm that this is the case.

## Methods

**Mouse dataset.** All in vivo experiments were conducted in accordance with the Italian law (DL 2006/2014, EU 63/2010, Ministero della Sanitá, Roma) and the recommendations in the Guide for the Care and Use of Laboratory Animals of the National Institutes of Health. Animal research protocols were reviewed and consented by the animal care committee of the Italian Institute of Technology and the Italian Ministry of Health. The rsfMRI dataset used in this work consists of $n = 19$ scans in adult male C57BL/6J mice that are publicly available[97,98]. Animal preparation, image data acquisition, and image data preprocessing for rsfMRI data have been described in greater detail

elsewhere[98]. Briefly, rsfMRI data were acquired on a 7.0-T scanner (Bruker BioSpin, Ettlingen) equipped with a BGA-9 gradient set, using a 72-mm birdcage transmit coil and a four-channel solenoid coil for signal reception. Single-shot BOLD echo planar imaging time series were acquired using an echo planar imaging sequence with the following parameters: repetition time/echo time, 1200/15 ms; flip angle, 30°; matrix, $100 \times 100$; field of view, $2 \times 2$ cm$^2$; 18 coronal slices; slice thickness, 0.50 mm; 1500 ($n = 19$) volumes; and a total rsfMRI acquisition time of 30 min. Mice were anesthetized with isoflurane (5% induction), intubated, and artificially ventilated (2% surgery). The left femoral artery was cannulated for continuous blood pressure monitoring and terminal arterial blood sampling. At the end of surgery, isoflurane was discontinued and substituted with halothane (0.75%).

Image preprocessing has been previously described in greater detail elsewhere[98]. Briefly, timeseries were despiked, motion corrected, skull stripped, and spatially registered to an in-house EPI-based mouse brain template. Denoising and motion correction strategies involved the regression of the mean ventricular signal plus 6 motion parameters. The resulting time series were band-pass filtered (0.01–0.1 Hz band) and then spatially smoothed with a Gaussian kernel of 0.5 mm full width at half maximum. After preprocessing, mean regional time series were extracted for 182 regions of interest (ROIs) derived from a predefined anatomical parcellation of the Allen Brain Institute (ABI[42,99]).

The mouse anatomical connectivity data used in this work were derived from a voxel-scale model of the mouse connectome made available by the Allen Brain Institute[100,101] (https://data.mendeley.com/datasets/dxtzpvv83k/2). Here, we preserved the directionality of connections—i.e., no symmetrization step was included in the pre-/post-processing pipelines.

Briefly, the mouse structural connectome was obtained from imaging-enhanced green fluorescent protein (eGFP)-labeled axonal projections derived from 428 viral microinjection experiments and registered to a common coordinate space[42]. Under the assumption that SC varies smoothly across major brain divisions, the connectivity at each voxel was modeled as a radial basis kernel-weighted average of the projection patterns of nearby injections[101]. Following the procedure outlined in[100], we re-parcelled the voxel scale model in the same 182 nodes used for the resting state fMRI data, and we adopted normalized connection density (NCD) for defining connectome edges, as this normalization has been shown to be less affected by regional volume than another absolute and/or relative measure of interregional connectivity[102].

**Human imaging dataset**. The Human Connectome Project (HCP) 3T dataset[33] consists of structural magnetic resonance imaging (T1w), functional magnetic resonance imaging (fMRI), and diffusion magnetic resonance imaging (dMRI) young adult subjects, some of which are twins. Here we use a subset of the available subjects. These subjects were selected as they comprise the 100 Unrelated Subjects released by the Connectome Coordination Facility. After excluding data based on completeness and quality control (4 exclusions based on excessive framewise displacement during scanning; 1 exclusion due to software failure), the final subset included 95 subjects (56% female, mean age = $29.29 \pm 3.66$, age range = 22–36). The study was approved by the Washington University Institutional Review Board and informed consent was obtained from all subjects.

A comprehensive description of the imaging parameters and image preprocessing can be found in[103]. Images were collected on a 3T Siemens Connectome Skyra with a 32-channel head coil. Subjects underwent two T1-weighted structural scans, which were averaged for each subject (TR = 2400 ms, TE = 2.14 ms, flip

angle = 8°, 0.7 mm isotropic voxel resolution). Subjects underwent four resting state fMRI scans over a 2-day span. The fMRI data was acquired with a gradient-echo planar imaging sequence (TR = 720 ms, TE = 33.1 ms, flip angle = 52°, 2 mm isotropic voxel resolution, multiband factor = 8). Each resting-state run duration was 14:33 min, with eyes open and instructions to fixate on a cross.

Finally, subjects underwent two diffusion MRI scans, which were acquired with a spin-echo planar imaging sequence (TR = 5520 ms, TE = 89.5 ms, flip angle = 78°, 1.25 mm isotropic voxel resolution, $b$-vales = 1000, 2000, 3000 s/mm$^2$, 90 diffusion weighed volumes for each shell, 18 $b = 0$ volumes). These two scans were taken with opposite phase encoding directions and averaged.

Structural, functional, and diffusion images were minimally preprocessed according to the description provided in[103] as implemented and shared by the Connectome Coordination Facility. Briefly, T1w images were aligned to MNI space before undergoing FreeSurfer's (version 5.3) cortical reconstruction workflow as part of the HCP Pipeline's PreFreeSurfer, FreeSurfer, and PostFreeSurfer steps. Functional images were corrected for gradient distortion, susceptibility distortion, and motion and then aligned to the corresponding T1w with one spline interpolation step. This volume was further corrected for intensity bias and normalized to a mean of 10,000. This volume was then projected to the 2 mm *32k_fs_LR* mesh, excluding outliers, and aligned to a common space using a multi-modal surface registration[104]. The resultant CIFTI file for each HCP subject used in this study followed the file naming pattern: *_Atlas_-MSMAll_hp2000_clean.dtseries.nii. These steps are performed as part of the HCP Pipeline's fMRIVolume and fMRISurface steps. Each minimally preprocessed fMRI was linearly detrended, band-pass filtered (0.008–0.008 Hz), confound regressed, and standardized using Nilearn's signal.clean function, which removes confounds orthogonally to the temporal filters. The confound regression strategy included six motion estimates, the mean signal from a white matter, cerebrospinal fluid, and whole brain mask, derivatives of these previous nine regressors, and squares of these 18 terms. Spike regressors were not applied. Following these preprocessing operations, the mean signal was taken at each time frame for each node, as defined by the Schaefer 400 parcellation[105] in *32k_fs_LR* space. Diffusion images were normalized to the mean b0 image, corrected for EPI, eddy current, and gradient non-linearity distortions, and motion, and aligned to the subject anatomical space using a boundary-based registration as part of the HCP pipeline's Diffusion Preprocessing step. In addition to HCP's minimal preprocessing, diffusion images were corrected for intensity non-uniformity with N4BiasFieldCorrection[106]. The Dipy toolbox (version 1.1)[107] was used to fit a multi-shell, multi-tissue constrained spherical deconvolution[108] to the data with a spherical harmonics order of 8, using tissue maps estimated with FSL's fast[109]. Tractography was performed using Dipy's Local Tracking module[107]. Multiple instances of probabilistic tractography were run per subject[110], varying the step size and maximum turning angle of the algorithm. Tractography was run at step sizes of 0.25, 0.4, 0.5, 0.6, and 0.75 mm with the maximum turning angle set to 20°. Additionally, tractography was run at maximum turning angles of 10°, 16°, 24°, and 30° with the step size set to 0.5 mm. For each instance of tractography, streamlines were randomly seeded three times within each voxel of a white matter mask, retained if longer than 10 mm and with valid endpoints, following Dipy's implementation of anatomically constrained tractography[111], and errant streamlines were filtered based on the cluster confidence index[112]. For each tractography instance, streamlined count between regions of interest were normalized by dividing the count between regions by the geometric average

volume of the regions. Since tractography was run nine times per subject, edge values were collapsed across runs. To do this, the weighted mean was taken with weights based on the proportion of total streamlines at that edge. This operation biases edge weights towards larger values, which reflect tractography instances better parameterized to estimate the geometry of each connection.

**Edge time series**. FC refers to the magnitude of statistical dependence between activity recorded from distant brain sites. In the present study, we define FC as the bivariate product-moment correlation. Consider regions $i$ and $j$ whose activity is denoted by the vectors $\mathbf{x}_i = [x_i(1), \ldots, x_i(T)]$ and $\mathbf{x}_j = [x_j(1), \ldots, x_j(T)]$. We can estimate their FC as $r_{ij} = \frac{1}{T-1} \sum_{t=1}^{T} z_i(t) \cdot z_j(t)$, where $z_i(t)$ and $z_j(t)$ represent the standardized ($z$-scored) regional time series.

Suppose we omitted the summation in calculating FC. Rather than the correlation coefficient $r_{ij}$, we would obtain the time series $r_{ij}(t) = [z_i(1) \cdot z_j(1), \ldots, z_i(T) \cdot z_j(T)]$. The elements of this time series have intuitive interpretations; they encode the magnitude, direction, and timing of co-fluctuations between regions $i$ and $j$. For instance, $r_{ij}(t) > 0$ if at time $t$ regions $i$ and $j$ both deflect in the same direction with respect to their means–i.e., $sign(z_i(t)) = sign(z_j(t))$. On the other hand, if $i$ and $j$ were deflecting in opposite directions, the $r_{ij}(t) < 0$. Relatedly, if at time $t$, the activity of $i$ and $j$ only slightly deviated from their respective means, the $|r_{ij}| \approx 0$. However, if the activity of either region deviates far from its mean, then $|r_{ij}| >> 0$.

Co-fluctuation time series have other useful properties. By design, they are an exact decomposition of a functional connection into its time-varying contributions. In previous studies, we found that most frames contribute little to the static connection weight. Rather, FC was driven by a select subset of high-amplitude frames. Across node pairs, these high-amplitude co-fluctuations tended to occur synchronously, giving rise to brain-wide high-amplitude events. In previous studies, we detected events by identifying frames where a measure of whole-brain co-fluctuation amplitude was statistically greater than that of a null model. Specifically, we calculated the root mean square (RMS) of all co-fluctuation time series: $\text{RMS}(t) = \sqrt{\frac{2}{N(N-1)} \sum_{i,j > i} r_{ij}(t)^2}$. From this time series, we identified its peaks—their amplitude and their timing. We then calculated $RMS$ time series using co-fluctuation time series estimated after circularly shifting the regional (parcel) time series. We repeated this procedure 1000 times, building up a null distribution of peak $RMS$ values, against which we compared the empirical values using non-parametric statistics. Events were defined as peaks in the intact co-fluctuation time series whose amplitude was statistically greater than that of the null distribution (false discovery rate fixed at 5% and critical $p$-value adjusted accordingly).

**Lin's concordance**. We measured the similarity between co-fluctuation patterns using Lin's concordance as opposed to the bivariate product-moment correlation. For two patterns with equal means and variances, these measures are identical. However, the concordance measure penalizes the similarity if the means differ from one another. For two vectorized co-fluctuation patterns $x$ and $y$, concordance is calculated as:

$$C_{xy} = \frac{2 \cdot Cov(x, y)}{Var(x) + Var(y) + (\mu_x - \mu_y)^2} \quad (1)$$

where $Cov(x, y)$ is the covariance, $Var(x)$ is the variance, and $\mu_x$ is the mean.

**Modularity heuristic**. Many networks exhibit mesoscale or community structure. This implies that they can be meaningfully decomposed into sub-networks referred to as modules or communities. The identity of these sub-networks is usually unknown ahead of time and cannot be determined from visual inspection alone, necessitating algorithmic approaches for estimating nodes' community assignments. The modularity heuristic is an objective function that maps a network and partition its nodes into non-overlapping communities to a scalar measure of quality, $Q$[43]. Intuitively, larger values of $Q$ are considered better partitions.

In more detail, $Q$ can be defined as:

$$Q = \sum_{ij} B_{ij} \delta(\sigma_i, \sigma_j) \quad (2)$$

where $B_{ij}$ is $\{i, j\}$ element of the modularity matrix, $B = W - P$, where $W$ is the observed connectivity matrix and $P$ is the connectivity matrix expected under a null model. The function $\delta(x, y)$ is the Kronecker delta and is equal to 1 when $x = y$ and 0 otherwise. The variable $\sigma_i$ denotes the community assignment of node $i$. In short, $Q$ is calculated as the sum of all within-community elements of the modularity matrix, $B$. It takes on a large value when the observed weights of those connections exceed their expected weights.

The modularity, $Q$, can also be expressed as a sum over communities. Given a partition of nodes into $K$ communities, we can write the contribution of community $c \in \{1, \ldots, K\}$ to the total modularity as $q_c = \sum_{i \in c, j \in c} B_{ij}$ such that $Q = \sum_c q_c$.

Previously we had described an algorithm for recursively applying modularity maximization to obtain a hierarchical partition of a network[18]. The algorithm works as follows. Given a fully weighted, symmetric, and possibly signed network, we denote its observed connectivity as $C$ and define its expected connectivity to be the mean of its upper triangle elements, $\langle C \rangle = \frac{2}{N(N-1)} \sum_{i,j > i} C_{ij}$. We can then define the modularity matrix, $B = C - \langle C \rangle$. Using the Louvain algorithm[113], we optimize $QN_{iter}$ times and perform consensus clustering on the ensemble of high-quality partitions[114] using a previously described algorithm[115]. Briefly, the consensus clustering procedure involves transforming the ensemble of partitions into a $N \times N$ coassignment matrix, whose elements count the fraction of times that every pair of nodes is assigned to the same community. We then calculate the expected coassignment matrix—i.e., how often we would expect nodes to be assigned to the same community given the same ensemble but where nodes are assigned to communities by chance. We then construct a consensus modularity matrix, $B_{consensus} = \text{Coassignment} - \text{Expected coassignment}$, and optimize the consensus modularity, $Q_{consensus} = \sum_{ij} B_{ij}^{consensus} \delta(\sigma_i^{consensus}, \sigma_j^{consensus})$. Because the coassignment matrix reinforces communities by mutually connecting *all* nodes assigned to the same community, it effectively reinforces consistently detected communities, making them more easily detectable when optimizing $Q^{consensus}$. Consequently, the variability across the ensemble of consensus partitions is typically much less than the variability across partitions in the initial ensemble. Frequently, the variability is zero—i.e., across all runs the Louvain algorithm converged to an identical solution. In that case, we regard the solution as the consensus partition. If the algorithm fails to reach a consensus, then we re-estimate the coassignment matrix and expected coassignment matrix given the variable estimates of consensus partitions, repeating this procedure until convergence. This step results in a consensus partition of nodes into $K$ communities and the contribution of each community to the total modularity, $q_c, c \in \{1, \ldots, K\}$.

Each of the $K$ communities can be viewed as a *child* of the *parent* network. To obtain a full multi-level and hierarchical description of the network's communities, we could recursively

apply the above procedure to all child networks and subsequently to the children of children and so on, until at the final level, every node is its own community. However, this procedure would be computationally expensive, especially for large networks. Moreover, many of the child networks may be poorly defined and not composed of cohesively connected nodes.

Accordingly, we introduce a statistical criterion that prunes branches from the hierarchy. Specifically, after obtaining the consensus partition, we permute consensus community labels to obtain a null distribution for communities' modularity contributions. Any community whose contribution was consistent with the null distribution was discarded and not subdivided further, effectively pruning its children from the hierarchical tree.

In this manuscript, we used the hierarchical algorithm both to partition static FC into communities at multiple resolutions as well as to cluster high-amplitude events into putative states.

In a previous study, we showed that co-fluctuation time series induces a bipartition of network nodes into two clusters[34]. One of the clusters corresponds to nodes with positive activations, while the other cluster corresponds to those with negative activations. The temporal average of co-assignment matrices obtained from these bipartitions was highly correlated with static FC.

When many co-fluctuation patterns are averaged together, as they are when we estimate event cluster centroids, estimating the bipartition is not straightforward. To obtain such an estimate we resort to data-driven algorithms. Namely, modularity maximization. Specifically, we define a modularity matrix by comparing the observed co-fluctuation matrix against a uniform null model. That is, $B_{ij} = W_{ij} - P$, where $P$ is a constant and is the same for all $\{i, j\}$. We optimize the corresponding modularity $N_{\text{iter}} = 1000$ times and use consensus clustering to obtain a single representation partition. From this partition, we extract the two largest clusters and, by inspection, ensure that they are anticorrelated with one another. We retain these two clusters as an estimate of the bipartition and discard any smaller clusters, assigning nodes in those clusters to a single non-cluster label.

Given an estimate of bipartition, we wanted to assess whether the two communities were also structurally segregated from one another—i.e., whether the bipartition was modular. To do so, we extracted the subgraph from the structural network—i.e., the connectome—composed of nodes in either of the two clusters. We also extracted the corresponding subgraph from the structural modularity matrix. Here, the modularity matrix was defined as $B_{ij}^{sc} = W_{ij}^{sc} - \frac{k_{i,\text{in}} k_{j,\text{out}}}{2m}$, where $k_{i,\text{in}} = \sum_j W_{ij}^{sc}$ weighted in-degree of node $i$ and $k_{i,\text{out}} = \sum_j W_{ji}^{sc}$ and $2m = \sum_i k_{i,\text{in}} = \sum_i k_{i,\text{out}}$.

Suppose that we let $c^+$ and $c^-$ correspond to the two clusters detected in the bipartition analysis and represent groups of nodes with high levels of positive and negative activity, respectively. Note that $c^+ \cap c^- = \{\emptyset\}$. Then, we can calculate the induced modularity of the bipartition as:

$$Q_{\text{induced}} = \sum_{i \in c^+, j \in c^+} B_{ij}^{sc} + \sum_{i \in c^-, j \in c^-} B_{ij}^{sc} \qquad (3)$$

To assess the statistical significance of the induced modularity, we compared the observed modularity against a null distribution generated using permutation tests. For two non-overlapping communities, $c^+$ and $c^-$, we generated a null distribution by sampling $n^+$ and $n^-$ at random and recomputing the induced modularity. Note that for the human imaging data, rather than using a random sample, we sampled communities using a spatially constrained spin test[46,47,116]. To do this, we defined a community vector of length $N$, where $N$ is the number of nodes. Elements of $c^+$ and $c^-$ were assigned values of 1 and 2, respectively, while all other elements were equal to 0. The spin test permutes this vector while approximately preserving the spatial contiguity of neural

elements. After spinning the vector, we extracted the subgraph corresponding to the non-zero elements in the community vector. Thus, this null model assesses whether subgraphs with similar spatial extent, equal size, and equal-sized communities could have generated a similar induced modularity.

**Statistics and reproducibility.** Statistical analyses were carried out using the full mouse dataset ($n = 19$ animals) and the 100 unrelated participants from the Human Connectome Project (of which $n = 95$ were included based on data quality control criteria). The primary claim of this paper is that event co-fluctuation patterns are underpinned by modular SC. We report this first using the mouse dataset and replicate it in the human dataset, despite differences in spatial scale, acquisition parameters, preprocessing pipeline, and modality used to reconstruct structural networks (tract tracing in mouse; tractography in human).

In Fig. 2b, c we compare different strategies for reconstructing FC. We compared strategy types using $t$-tests. In Fig. 4c, f, I, we compare observed modularity scores, $Q$ values, against two null distributions, each estimated using permutation tests. For each comparison we calculated a non-parametric $p$-value as the fraction number of permutations that were at least as large as the observed $Q$ value. We used the same procedure to estimate $p$-values for the data displayed in Fig. 5d, g, j.

**Reporting summary.** Further information on research design is available in the Nature Portfolio Reporting Summary linked to this article.

## Data availability

Human Connectome Project data are available from https://db.humanconnectome.org/ after signing a data use agreement. Mouse data have been deposited and are available https://data.mendeley.com/datasets/7y6xr753g4/1, https://data.mendeley.com/datasets/r2w865c959/1, and https://data.mendeley.com/datasets/thpszcwcgx/2. The data used to make the plots in Figs. 2–5 can be found in Supplementary Data 1–4, respectively.

## Code availability

The code for detecting events is available here (https://github.com/brain-networks/event_detection).

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

## Acknowledgements

E.R. was supported by the Oberlin College Junior Practicum program. R.F.B. acknowledges support from the National Science Foundation (NCS-FO award #2023985). J.F. acknowledges support from the National Institutes of Mental Health (1ZIAMH002783-20).

## Author contributions

E.R. and R.B. conceived the project, carried out analyses, and wrote the first draft of the paper. L.C. and A.G. contributed mouse functional imaging and parcellated structural connectivity data. J.F. processed human imaging data. E.R., J.T., Y.J., F.Z.E., J.F., M.P., L.C., A.G., and R.B. wrote and edited the paper.

## Competing interests

The authors declare no competing interests.
