## [Peer Review File · Communications Biology]

Reviewers' comments:

Reviewer #1 (Remarks to the Author):

In this elegant report, Ragone et al. conducted an intriguing study investigating co-fluctuation in mouse and human brains. The analysis is commendable, and the study is well-written, with conclusions well-supported by the results. I have only a few suggestions.

In the introduction, the authors stated that "These activity patterns are constrained by the brain's underlying structural connectivity (SC)." It would be beneficial if they could address recent findings from Fornito's group (<https://www.nature.com/articles/s41586-023-06098-1>) regarding the geometrical role of the brain in generating functional connectivity (FC). This is a hot and somewhat controversial topic in connectivity research, and discussing these contrasting viewpoints could enrich the study.

Additionally, it would be valuable to mention and compare the iCAPs/CAPs methodology, which has been recently applied in mouse models as well (e.g., <https://www.nature.com/articles/s41598-023-36812-y>) and is similar to the applied methodology

While I enjoyed reading the methodological preprocessing part, it would be helpful to introduce the specific aim for each analysis to assist readers in navigating the complex analytical approach. For instance, when referring to Figure 1 panel d, it took me some time to understand the 'event cutoff,' which I later realized was related to peaks greater than a null distribution. Clarifying such details would enhance the paper's readability. Lastly, I was unable to find the results from the consensus clustering in the report.

Reviewer #2 (Remarks to the Author):

This is an excellent manuscript examining the occurrence and organization of functional spontaneous co-fluctuations in brain function as estimated using fMRI. The authors have used data from anesthetized mice and the Human Connectome project to conduct this work. The anesthetized mice represent a high-quality dataset, that perhaps needs a bit more context in this paper. How do we reconcile not just the species but also the levels of consciousness in the animal models. Are there physiological patterns in both the human and animal work that can be added to the framework?

In addition, I applaud the overall effort for combining human and animal data in the same manuscript. But ... I do believe that the readers need more handholding. I don't see the organizational features that are common to the species. To this end, I would urge the authors to find an improved representation of the dataset.

How quality parameters influence findings needs a bit more treatment as well.

Reviewer #3 (Remarks to the Author):

The authors presented a brain imaging study in mice demonstrating the presence of co-fluctuation events and their relation to functional and structural subgraph modules. They replicated some of the results in a human fMRI cohort suggesting cross-species consistency of the phenomenon. I welcome the overall idea of leveraging rodent fMRI to allow for translational research. The presence of high-

amplitude co-fluctuation events and the structural basis of the functional models appears to be convincing, but I was not able to fully grasp the main analysis steps given the wealth of analyses described and unclear presentation.

My main concern is that the manuscript as such is lacking focus and it is difficult to parse for someone with an interest in computational neuroscience, without being an expert in network modelling. In 5 main figures and 10 supplementary figures, a large amount of results is shown, some of which seem to represent technical nuances that are distracting from the main flow and main argument. The results section is at times referring to high-level results and at other times detailing technical considerations, which leaves the impression of a technical report. Because of these main concerns, I encourage the authors to re-submit a *substantially* revised and rewritten version of the manuscript. I do not recommend publication of the paper as it is.

These are also some minor points:

- overall, there are quite a few plots with missing labels, abbreviations are not introduced, typos, etc.

Abstract and Introduction:

- The statement 'no studies have applied edge-centric frameworks to study fMRI in model organisms' is not correct (see for example Luppi 2022 Nat Neurosci) and it is further not clear what is meant by the 'same framework'. This sentence should be toned down appropriately. In the discussion they acknowledge that this is 'one of the first' studies to apply this framework.

- What is meant by 'clusters': brain areas or temporal clusters? What is a 'centroid', which method was used to determine 'axonal' projections?

- What is meant by fine-scale investigations? fMRI in rodents?

Results and Methods:

- what type of anaesthesia was used? It seems like an important detail to mention for the cross-species comparison.

- Some methodological considerations are repeated or introduced in depth in the results (for example the estimation of bipartition), whilst other critical information is omitted, for example how was static FC computed?

- Figure 1 seems to be missing some information: E: should the label be Edge-pairs rather than edges? F: what are the different boxes? Different time points? What figure represents the RMS timeseries?

- Is static FC definition part of this figure as Figure 2 is referenced only later in the text?

- Figure 2: What are the axis labels for the scatter plots?

- Figure 3: what is EV?

- Figure 4: Could the hippocampus be annotated in the graph? What does the size of each node in the dot-plot suggest?

- Figure S2: Could the abbreviations be introduced?

- Figure S7: What does the abbreviation Ind. stand for?

- the comparison of axonal tracing and dMRI tractography in humans bears limitations that should be discussed.

We thank all reviewers for their commitment to helping us improve this submission. In responding to their comments, we believe that our enclosed submission is much improved.

Throughout this response letter we refer to comments made by the reviewers in black text. Our responses are shown in **blue text**. Changes to the manuscript are depicted in *italicized red text* to highlight the specific edits.

When appropriate, we note the section name where changes were implemented. In addition to this response letter, we include two versions of the revised manuscript: a “marked” version that tracks all of the changes made since the original submission and a “clean” version that includes all changes but without any highlights.

Rick Betzel

Reviewer #1 (Remarks to the Author):

In this elegant report, Ragone et al. conducted an intriguing study investigating co-fluctuation in mouse and human brains. The analysis is commendable, and the study is well-written, with conclusions well-supported by the results. I have only a few suggestions.

We appreciate the authors overall positive evaluation and are happy to address their additional concerns.

In the introduction, the authors stated that "These activity patterns are constrained by the brain's underlying structural connectivity (SC)." It would be beneficial if they could address recent findings from Fornito's group (<https://www.nature.com/articles/s41586-023-06098-1>) regarding the geometrical role of the brain in generating functional connectivity (FC). This is a hot and somewhat controversial topic in connectivity research, and discussing these contrasting viewpoints could enrich the study.

This is an interesting point and one we are happy to consider. In the paper by Pang et al (2023), the authors examine a mode-based model of brain activity. Their model posits that brain activity emerges from vibrations of structural eigenmodes. By selectively reweighting these modes in different proportions, the authors show that they can produce different patterns of activation.

The model itself, is not inconsistent with the notion that SC shapes brain activity. Indeed, eigenmodes of structural connectomes do a good job of explaining brain activity and FC (see, for example, Atasoy et al 2016 Nat. Comm/ or Preti et al 2023 ICASSP, IEEE proceedings).

However, eigenmodes can be derived from any connectivity matrix, including the triangular surface mesh. Unlike white-matter connections, surface mesh connectivity is regular and repeating, reflecting much shorter connections.

Where the paper by Pang et al becomes controversial, is in their finding that on a per-mode basis, eigenmodes from the geometric mesh do a better job explaining task contrast maps and spontaneous activity than eigenmodes derived from long-distance white-matter.

On one hand, this observation conforms with the extant literature. A massive number of studies have shown that, at the meso- and macro-scales, connectomes are dominated by short-range connections. That is, the probability of two brain areas being connected falls off with distance, as do the weights of those long-distance connections. Coupled with the observation that fMRI BOLD contrast maps are generally spatially autocorrelated and given the fact that Pang et al use up to 200 modes to model these maps, the performance of the geometric maps might even be unsurprising.

In addition, there are other concerning aspects of that paper. Namely, the margin by which the geometric modes outperformed the white-matter modes is relatively small. While it is unclear if

alternative processing pipelines for dMRI or could make up this difference, the comparable performance of these two modalities suggests that claims, like those made by Pang et al wherein “geometry is a fundamental constraint” on brain function, are likely overstated.

Additional methodological concerns have been raised publicly, with Faskowitz et al (2023) noting that geometric brain modes predict random (spatially correlated) brain maps, and that geometric modes derived from random spheroid surfaces predict empirical brain maps. That is, the methods used by Pang et al to model brain function in terms of modes exhibit low specificity. Others have been more measured in their responses, with Luo et al (2023) suggesting that additional experimental evidence is necessary before the results of Pang et al can be embraced.

In short, the modeling framework by Pang et al is not inconsistent with the notion that brain activity (and functional connectivity) are underpinned by structural connectivity. Additionally, the controversial claim by Pang et al – that geometry and not structural connectivity underpins brain function – has serious flaws, both technically and conceptually.

Nonetheless, we agree that given the high-profile nature of the Pang et al (2023) paper, it is beneficial to re-examine the premises of our submission as well as our results through the lens of Pang et al. We now include a brief discussion of how our findings might relate to their observations alongside a candid discussion of the limitations of the Pang et al paper.

The revised text reads:

“Our findings suggest that modular structure of long-distance anatomical connectivity helps shape the organization of high-amplitude co-fluctuation patterns. Our analyses are motivated by a century of neuroscientific observations, demonstrating that perturbations to long-distance connectivity result in acute changes to function [74,75]. However, a recent study has challenged this very premise, positing that long-distance connections play only a small role in shaping brain function [76]. Rather, brain function emerges from short-range, regular connectivity patterns that reflect the brain's intrinsic geometry. While this perspective has yet to be fully embraced by the neuroscientific community [77,78], high-amplitude events could serve as another feature along which to compare and adjudicate between the two theories. That is, given the apparent ubiquity of events in large-scale imaging data, it would be interesting to assess whether events appear in simulated time series generated by the geometric models and, if so, to explore their structural underpinnings. We leave this exploration for future studies.”

Additionally, it would be valuable to mention and compare the iCAPs/CAPs methodology, which has been recently applied in mouse models as well (e.g., <https://www.nature.com/articles/s41598-023-36812-y>) and is similar to the applied methodology

This is an important point. We mentioned the deep mathematical relationship between network-level events detected with edge time series and the iCAPs/CAPs patterns, and we sincerely appreciate the opportunity to elaborate on it.

For reference, the original text discussing the links between CAPs and co-fluctuation patterns is reprinted below:

“We also examined the activity modes that underlie each of these clusters. Interestingly, they bore a striking resemblance to the results of a recent analysis of high-amplitude mouse activations, in which the authors detected six co-activation patterns (CAPs) that were present in both awake and anesthetized mice [51]. These patterns could be grouped into anti-correlated pairs, such that the spatial patterns of CAPs that correspond to a given pair are approximately anticorrelated with one another.

Notably, the reported CAPs resembled the activity modes that underpinned the event cluster centroids described here (CAPs 3 and 4, 1 and 2, and 5 and 6 mapped onto clusters 1-3, respectively). This is not a coincidence; mathematically, co-fluctuation matrices are calculated as the product of an activation vector with itself transposed. Consequently, an activation pattern (or the same pattern where the sign of each element is flipped) would generate an identical co-fluctuation matrix [30]. Hence, events can be viewed as a connectivity-based analog of the activation-centric CAPs, and likely explain the parallels between results presented here and in other studies that analyzed the same dataset using CAPs [51,52].”

We have extended the text in the final paragraph to further describe the link between CAPs and events and have cited, among other papers, the reference recommended by the reviewer.

The revised text reads:

*“Hence, events can be viewed as a connectivity-based analog of the activation-centric CAPs, and likely explain the parallels between results presented here and in other studies that analyzed **“this same dataset [51,52] and others [53-55] using CAPs. In fact, we speculate that in most studies that report CAPs, there will be a two-to-one mapping of CAPs to event co-fluctuation patterns. Both are detected as frames with exceptionally high-amplitude activity (CAPs) or co-fluctuations (edge time series), and the mathematical relationship between activations and edges stipulate that every co-fluctuation pattern can arise from two perfectly anti-correlated activation maps.”***

While I enjoyed reading the methodological preprocessing part, it would be helpful to introduce the specific aim for each analysis to assist readers in navigating the complex analytical approach. For instance, when referring to Figure 1 panel d, it took me some time to understand the 'event cutoff,' which I later realized was related to peaks greater than a null distribution.

Clarifying such details would enhance the paper's readability. Lastly, I was unable to find the results from the consensus clustering in the report.

We apologize for any confusion or omissions. We have carefully read through the entire text, liberally updating sections that we flagged as possibly lacking experimental/methodological details or where those details were included only in a reference. These include descriptions of the consensus clustering approach.

Below, we include all updated text. In particular, we focus on the consensus clustering methods, which were not clearly described in our initial submission.

The revised description reads:

*Previously we had described an algorithm for recursively applying modularity maximization to obtain a hierarchical partition of a network [18]. The algorithm works as follows. Given a fully-weighted, symmetric, and possibly signed network, we denote its observed connectivity as C and define its expected connectivity to be the mean of its upper triangle elements, $\langle C \rangle = \frac{2}{N(N-1)} \sum_{i,j>i} C_{ij}$. We can then define the modularity matrix, $B = C - \langle C \rangle$. Using the Louvain algorithm [101], we optimize Q N_{iter} times and perform consensus clustering on the ensemble of high-quality partitions [102] using a previously described algorithm [103]. **Briefly, the consensus clustering procedure involves transforming the ensemble of partitions into a $N \times N$ coassignment matrix, whose elements count the fraction of times that every pair of nodes is assigned to the same community. We then calculate the expected coassignment matrix--i.e. how often we would expect nodes to be assigned to the same community given the same ensemble but where nodes are assigned to communities by chance. We then construct a consensus modularity matrix, $B_{consensus} = Coassignment - Expected Coassignment$, and optimize the consensus modularity, $Q_{consensus} = \sum_{ij} B_{ij}^{consensus} \delta(\sigma_i^{consensus}, \sigma_j^{consensus})$. Because the coassignment matrix reinforces communities by mutually connecting all nodes assigned to the same community, it effectively reinforces consistently detected communities, making them more easily detectable when optimizing $Q_{consensus}$. Consequently, the variability across the ensemble of consensus partitions is typically much less than the variability across partitions in the initial ensemble. Frequently, the variability is zero--i.e. across all runs the Louvain algorithm converged to an identical solution. In that case, we regard the solution as the "consensus partition". If the algorithm fails to reach consensus, then we re-estimate the coassignment matrix and expected coassignment matrix given the variable estimates of consensus partitions, repeating this procedure until convergence. This step results in a **consensus** partition of nodes into K communities and the contribution of each community to the total modularity, $q_c, c \in \{1, \dots, K\}$.***

We have also updated caption under Figure 1. The updated text provides a more complete

(though still high-level) description of the complete procedure for transforming activation time series into edge time series, detecting events, and for clustering them into putative states.

We copy, here, the updated figure and text:

Figure 1. Schematic illustrating edge time series construction and clustering. (a) Array of parcel time series. Rows and columns correspond to parcels (ordered by anatomical division) and frames, respectively. (b) Whole-brain functional connectivity is typically estimated as the correlation matrix of parcel time series. That is, the weight of the functional connection between nodes i and j is specified as the product-moment correlation coefficient, r_{ij} . (c) The procedure for estimating r_{ij} entails z-scoring each parcel time series, calculating their elementwise product—i.e. $z_i(1) \times z_j(1), \dots, z_i(T) \times z_j(T)$ —and taking the mean of those products (sum divided by the number of samples divided by one). Omitting the averaging step yields the co-fluctuation (or “edge”) time series $r_{ij}(t) = [z_i(1) \times z_j(1), \dots, z_i(T) \times z_j(T)]$, whose elements encode time-varying changes in the weight of the connection between nodes i and j . In this panel, we show time series for regions i and j (green and blue curves) and their corresponding edge time series (gray). (d) We can calculate edge time series for all node pairs (edges) in the network and arrange them as rows in an edge-by-frame matrix. (e) Previous studies identified infrequent and short-lived high-amplitude bursts. These bursts or “events” can be detected by calculating the root mean square (RMS) across all edges at each frame and identifying peaks whose amplitude exceeds that of a null distribution (estimated using same procedures as empirical RMS, but starting with circularly shifted parcel time series). (f) The co-fluctuation patterns expressed during events are very different than those expressed during low-amplitude frames. Here, we highlight approximately 175 frames and show whole-brain co-fluctuation matrices for three local maxima (events; red border) and three local minima (troughs; gray border). (g) Although no two events are identical in terms of co-fluctuation patterns, events can be grouped (broadly) into clusters. We detect them by computing the similarity (concordance) between all pairs of events and directly clustering the resulting matrix using a hierarchical algorithm. Here, we highlight two large clusters and their respective centroids (the mean co-fluctuation pattern across all events assigned to each cluster).

Additionally, and in line with the reviewer’s suggestion to enhance readability by clarifying methodological details, we have updated both our **Results** and **Materials and Methods**. We identified several sections where details were not clearly presented and now include them in the description. See below for updates to the text.

In the subsection “High-amplitude co-fluctuations recapitulate static FC”:

“One of the first observations made using edge time series was that, with only a small subset of high-amplitude frames--putative “events”--it was possible to accurately reconstruct static FC [13]. Note that here we define whole-brain FC as the matrix of all interregional correlations. To date, these types of analyses have been carried out largely using functional imaging data acquired from awake humans; whether such events exist in data acquired from animals has been underexplored. Here, we assess whether a similar effect is evident when we apply edge time series to functional imaging data acquired from anesthetized mice.

Our procedure for testing this hypothesis included a series of post-processing analysis steps. First, we transformed fMRI BOLD time series from $N=182$ parcels (Fig. 1a) into edge time series. This procedure involved standardizing (z-scoring) each time series and, for each of the $N(N-1)/2=16471$ pairs, calculating their framewise product (Fig. 1c). The result is a co-fluctuation or “edge time series” for every pair of nodes whose elements encode the timing, amplitude, and sign of interregional co-fluctuations. Notably, the temporal average of a given edge time series is exactly the bivariate product-moment correlation--i.e. FC (Fig. 1b). Thus, edge time series can be viewed as exact decompositions of static FC into time-varying (framewise) contributions.”

To both summarize these findings and to form a logical bridge between this section and the next, we have added the following paragraph:

Collectively, these findings suggest that a small fraction of high-amplitude frames explain the structure of time-averaged, static FC. Importantly, these findings extend previous lines of research linking events and FC in human brains to mouse models. In the following section, we explore the spatial structure of mouse events in greater detail.

In the section entitled “High-amplitude events can be sub-divided into recurring network states”, we have added the following justification to the first paragraph:

“Previous studies have shown that high-amplitude and network-level events can be clustered into putative states on the basis of their topographic similarity to one another [15,18,19]. It is unclear whether the same is true for the mouse edge time series data analyzed here. Further, if events can be grouped into clusters, the features that distinguish events in one cluster from those in another are unknown.”

We have also made several additional edits to help clarify methodological details:

“To address these questions, we followed the analysis pipeline from Betzel et al (2022). Briefly, this involved aggregating event co-fluctuation patterns across all subjects, calculating the similarity (Lin's concordance) between all pairs of events (Fig. 3a), and recursively applying modularity maximization (coupled with consensus clustering and a

statistical criterion for terminating the recursion) to obtain a hierarchy of statistically significant event clusters (Fig. 3b,c). Note that for two co-fluctuation patterns with identical means and variances, Lin's concordance resolves to the familiar correlation similarity metric. However, unlike correlation, Lin's concordance decreases as the difference between means and variances grows (see **Materials and Methods**)."

And added a summary paragraph:

"Collectively, and like the results described in the previous sub-section, these findings closely align with analyses of events detected in human functional imaging data. Namely, we show that event co-fluctuation patterns can be described with a relatively small number of clusters, hinting at an approximately finite and discrete repertoire of co-fluctuation states. An important open question, however, is how these patterns emerge from the static underlying anatomical connectivity--i.e. the connectome. We explore this question in the following section."

We also made edits to the **Results** sub-section "High-amplitude co-fluctuation patterns reflect modular sub-divisions of mouse anatomical connectivity".

Some of the edits helped with the overall narrative:

"One of the most important questions in neuroscience is how brain structure relates to brain function. This question can be reframed in the context of this present study: how does the static scaffolding of anatomical connectivity imprint upon measures of brain function--i.e. event co-fluctuation patterns? In Pope et al (2021), the authors detected events in synthetic fMRI BOLD data generated by an anatomically constrained oscillator model. ..."

Other edits were made to help specify details of the analysis methods:

*"Our strategy for comparing event co-fluctuations and anatomical connectivity deviated from that of Pope et al (2012), which depended upon a specific definition of anatomical modules. Instead, our approach was to recover the bipartition of network nodes into positively and negatively fluctuating groups associated with each event cluster centroid [30]. If we were to examine a single co-fluctuation pattern, its bipartition is defined unambiguously. However, for event cluster centroids, which reflect the mean over many co-fluctuation patterns, recovering the bipartition is not as straightforward and requires additional analysis. One possible solution is to apply clustering algorithms--e.g. modularity maximization--to the centroid networks (see **Materials and Methods**). This procedure is not guaranteed to return a perfect bipartition--i.e. exactly two communities--but we can heuristically treat the two largest and anticorrelated modules as estimates of the bipartition. Any other communities, which by definition are smaller, are considered peripheral and not included in subsequent analyses."*

“Given an estimate of the bipartition, we then imposed this partition onto the network of structural connections and calculated the modularity that it induced [41]. That is, we calculated the extent to which structural connections concentrate within communities compared to chance. We compared the observed modularity against two null models; one in which we randomly assign nodes to either group, destroying spatial autocorrelations, and another in which we approximately preserve the variogram--i.e. the spatial dependencies--of the original data [42].”

“For event cluster one, these results were not dependent on the resolution parameter used to define the cluster--a free parameter that controls the number and size of detected communities.”

We also made edits in the final **Results** sub-section, entitled “Replicating event-module relationships using human imaging data”.

“One of the aims of this study was to replicate using mouse imaging data, several key findings that have already been made using human data. In the previous section, however, we identified a novel link between structural connectivity and event co-fluctuations using mouse imaging data. In this section, we reverse our course, and seek to replicate this finding in human data. Unlike the mouse connectome, the human connectome is typically reconstructed non-invasively from tractography and diffusion MRI. ...”

Finally, we made edits to the **Materials and Methods** section. We show those edits below.

“Functional connectivity (FC) refers to the magnitude of statistical dependence between activity recorded from distant brain sites. In the present study, we define FC as the bivariate product-moment correlation. Consider regions i and j ...”

Reviewer #2 (Remarks to the Author):

This is an excellent manuscript examining the occurrence and organization of functional spontaneous co-fluctuations in brain function as estimated using fMRI.

We appreciate the positive remarks from the reviewer.

The authors have used data from anesthetized mice and the Human Connectome project to conduct this work. The anesthetized mice represent a high-quality dataset, that perhaps needs a bit more context in this paper. How do we reconcile not just the species but also the levels of consciousness in the animal models. Are there physiological patterns in both the human and animal work that can be added to the framework?

This is an excellent point that we agree was not adequately addressed in our initial submission.

As noted in the main text, prior studies have shown that events are shaped by (at least) two distinct factors: their timing is modulated by sensory input (see Zamani Esfahlani et al 2020; Tanner et al 2022; Levakov et al 2023) and their spatial topography is shaped by the modular structure of the underlying anatomical network (see Pope et al 2021).

Given that the mice were anesthetized, it is unlikely that the detected events are driven by sensory/perceptual signals. Rather, the events detected in mice are more likely the product of stochastic, spontaneous, and anatomically constrained fluctuations (a hypothesis supported by a recent preprint; Fasoli et al 2022). In the human dataset, where participants are awake but at rest, events are likely a combination of both factors, as well as others that have yet to be elucidated.

Besides the likely shared contributions from structural connectivity, there are other possible explanations for why we observe cross-species convergence, despite differences in level of consciousness. One possibility outlined by Laumann & Snyder (2021) is that spontaneous activity (or in our case, co-activity) might reflect mechanisms of synaptic plasticity. That is, co-activations of regions might correspond to Hebbian signaling (the principle of “fire together, wire together”), consolidating brain regions into cohesive subnetworks. Spontaneous and co-activity could also reflect homeostatic processes, adjusting regional excitatory/inhibitory balance to return it to some working point. Notably, both of these processes can occur “offline”, i.e. during sleep, loss of consciousness, etc.

We have rewritten the Discussion sub-section “Events occur in absence of consciousness”. The revised text reads:

“Here, we show that events in both mouse and human functional imaging data are undergirded by modular structural networks. This cross-species convergence is particularly interesting given differing levels of consciousness--mice were anesthetized

while human participants were awake. Although previous studies have suggested that event timing is modulated by sensory inputs (Zamani Esfahlani et al 2020; Tanner et al 2022; Levakov et al 2023), this observation indicates that consciousness is not a requirement for the emergence of events. Broadly, this observation is aligned with the work of Pope et al (2021), who demonstrated that events can occur stochastically--i.e. in the absence of exogenous input--if the structural network is sufficiently modular.

On the other hand, these results raise the question: exactly what is the purpose of events? Clearly, they have both online and offline contributions. Focusing on the offline component, one possibility is that spontaneous co-fluctuations (including events) reflect Hebbian tuning of synaptic weights, reinforcing network modules (Laumann & Snyder 2021). Another possibility, also outlined by Laumann & Snyder (2021) is that spontaneous co-fluctuations have a restorative, homeostatic effect and return neuronal populations towards excitatory/inhibitory balance. Notably, these processes occur offline and could therefore also explain the observation of events in the brains of anesthetized mice.

More broadly, these observations underscore the need to more clearly elucidate not only the network-level mechanisms that support the emergence of events, but the contributions of neurobiological, physiological, molecular, and environmental factors, as well. Addressing this question requires challenging cross-disciplinary and multi-scale research but could shed light on the role(s) of events in normative brain function.”

In addition, I applaud the overall effort for combining human and animal data in the same manuscript. But ... I do believe that the readers need more handholding. I don't see the organizational features that are common to the species. To this end, I would urge the authors to find an improved representation of the dataset.

We apologize for the confusion and welcome the opportunity to clarify. From the outset, the aim of our paper was to replicate, using mouse imaging data, results that had been previously reported in human imaging data. Namely, that events recapitulate static FC and that there are a small set of approximately repeating event clusters.

Using the mouse data, we also made a novel observation. Specifically, we bipartitioned the mouse brain into two subgraphs based on the event cluster co-fluctuation patterns. We then asked, if we imposed this partition on the structural connectome, would the partition be modular? That is, would the within-subgraph connections be stronger than expected under a null model? We found that this was, indeed, the case. It suggests that the high-amplitude co-fluctuation patterns (events) have structural underpinnings.

This observation was novel. It had never been made in mouse *or* human brains. We therefore tested whether there was a similar effect in the human imaging dataset. We found that this was the case. Bipartitions were highly modular; much more than chance.

The salient figure panels that demonstrate these effects are Figure 4c,f,i (induced modularity for mouse) and Figure 5d,g,j (induced modularity for humans). The aim is not to draw comparisons between the co-fluctuation patterns themselves, but their relationships to structural connectivity.

How quality parameters influence findings needs a bit more treatment as well.

This is an important point and one we thank the reviewer for noting. The link between functional connectivity and data quality (especially as indexed by measures of in-scanner motion), has been well-documented.

One of the obvious concerns, especially with analyses of “events,” which are essentially brainwide spikes in neural recordings, is that they might reflect or be impacted by in-scanner motion. Because the mice were anesthetized, their motion was much lower than that of a typical human resting fMRI scan (mean \pm s.d. framewise displacement of 0.031 ± 0.015). Nonetheless, there are motion spikes. In a subsequent reanalysis of data, we performed the following steps:

1. Identified and excluded “high-motion frames” that exhibited framewise displacements of 0.075 mm or greater.
2. Further excluded frames that were within 5 frames of a “high motion frame”.
3. Excluded sequences of the remaining, low-motion frames if their total length (number of frames) was less than 10.

This procedure resulted in discarding, on average, 47.5 frames per scan (approximately 3% of all frames). One mouse exhibited excessive motion, which resulted in discarding 417 frames (approximately 27%). Excluding this mouse, the number of discarded frames was between 0 and 78 (approximately 5%).

Using these low-motion frames, we performed event detection as described in the main text. We actually identified a greater number of events than in the main text (641 as opposed to 624). This difference reflects, in part, the stochasticity of our event detection algorithm—our null RMS distribution gets generated using a finite set of samples and is therefore somewhat variable from run to run. On the other hand, if we remove motion spikes (which can exhibit high amplitudes), we likely obtain a null distribution with fewer extreme points, making it possible for frames with lower overall amplitude to be classified as an event.

We then clustered these frames using the hierarchical analog of modularity maximization. As in the original submission, we found evidence of three large clusters accounting for 39.5%, 31.2%, and 8.0% of all events. We identified the centroids of these clusters as the mean co-fluctuation pattern across all events assigned to each cluster. Their spatial similarity (correlation coefficient) to the three event clusters described in the main text were $r_1 = 0.998$, $r_2 = 0.996$, and $r_3 = 0.96$ (here, subscript denotes the cluster label).

In other words, these three clusters, which we analyzed in-depth in the original submission, are almost unchanged following a motion censoring procedure that excluded high-motion frames. Because these cluster centroids are propagated to the next level of analysis (linking event clusters to anatomical connectivity) and because their similarity to the centroids reported in the main text is near exact, we anticipate that the induced modularity analysis will return nearly identical results.

We now include the details of this analysis as a supplementary figure. For completeness, we copy and include the corresponding figure and its caption below:

Figure S7. **Assessing impact of in-scanner motion on event clusters.** In the main text we detected high-amplitude events, which we subsequently clustered and linked to the modular structure of anatomical connectivity. Although the fMRI data were processed using a pipeline that include denoising and motion correction steps, it is possible that residual motion impacted event detection so that motion spikes were coincident with event timing (though note that the mouse data exhibit a relatively low level of in-scanner motion from the outset; mean \pm s.d. framewise displacement of 0.031 ± 0.015). To address this concern, we repeated event detection and clustering after including a more conservative motion screening procedure. Briefly, we excluded from analysis any frame whose framewise displacement exceeded a threshold of 0.075 mm. We also excluded frames within five samples of a high-motion frames and dropped any low-motion frames that were part of short contiguous sequences (fewer than ten low-motion frames in series). With the exception of one animal for whom 417 frames (approximately 27%) were flagged as high-motion, the mean number of frames discarded was 47.5 (range of 0 to 78 frames). This event detection procedure identified 641 events as opposed the 624 originally reported. The hierarchical clustering procedure identified three large clusters at the second level, comprising 39.5%, 31.2%, and 8.0% of all events. These clusters, whose centroids shown above in the bottom row next to the label “Low-motion” exhibited a near perfect correspondence with the clusters reported in the main text (labeled “No correction”). These cluster centroids get propagated to the next analysis, in which they are linked to anatomical connectivity. Because of their near-perfect correspondence with the originally reported cluster centroids, they will induce almost identical levels of modularity, replicating the results in the main text.

Reviewer #3 (Remarks to the Author):

The authors presented a brain imaging study in mice demonstrating the presence of co-fluctuation events and their relation to functional and structural subgraph modules. They replicated some of the results in a human fMRI cohort suggesting cross-species consistency of the phenomenon. I welcome the overall idea of leveraging rodent fMRI to allow for translational research. The presence of high-amplitude co-fluctuation events and the structural basis of the functional models appears to be convincing, but I was not able to fully grasp the main analysis steps given the wealth of analyses described and unclear presentation.

We thank the reviewer for their honest evaluation and for the opportunity to clarify the aims of our analysis through our detailed responses

My main concern is that the manuscript as such is lacking focus and it is difficult to parse for someone with an interest in computational neuroscience, without being an expert in network modelling. In 5 main figures and 10 supplementary figures, a large amount of results is shown, some of which seem to represent technical nuances that are distracting from the main flow and main argument. The results section is at times referring to high-level results and at other times detailing technical considerations, which leaves the impression of a technical report. Because of these main concerns, I encourage the authors to re-submit a **substantially** revised and rewritten version of the manuscript. I do not recommend publication of the paper as it is.

These are also some minor points:

- overall, there are quite a few plots with missing labels, abbreviations are not introduced, typos, etc.

We thank the reviewer for the careful read and, in replying to the rest of their queries/comments, believe we have addressed these issues.

Abstract and Introduction:

- The statement 'no studies have applied edge-centric frameworks to study fMRI in model organisms' is not correct (see for example Luppi 2022 Nat Neurosci) and it is further not clear what is meant by the 'same framework'. This sentence should be toned down appropriately. In the discussion they acknowledge that this is 'one of the first' studies to apply this framework.

We appreciate the reviewer's perspective and for the opportunity to clarify.

In our study, we adopt an "edge-centric" approach. This refers to a specific set of tools and measurements that were outlined in Zamani Esfahlani et al (2020) PNAS and Faskowitz et al (2020) Nature Neuroscience. In those papers, the authors show that when FC is defined as a correlation coefficient, which is common in human neuroimaging, the coefficient can be decomposed into its time-varying contributions. This procedure yields an "edge time series" for every pair of nodes. In fact, this is where the term "edge-centric" originates; because the

procedure described by Faskowitz et al and Zamani Esfahlani transforms regional time series into edge time series, we can think of it as shifting emphasis from nodes and onto edges. Hence the term “edge-centric”.

To our knowledge, there exists only one application of edge time series to non-human data (the paper by Rabuffo et al 2021, which we cite). However, rather than analyze empirical data, it builds dynamic and generative models of mouse activity (and from those, eventually derives edge time series).

The paper referenced by the reviewer (Luppi et al 2022) uses an entirely different approach from the one outlined above. Namely, they use an information-theoretic decomposition to generate two version of static FC—one where connections’ weights are based on “synergistic” relationships between regional time series and another where they are based on “redundant” relationships. Their approach does not generate time-resolved networks in the spirit of Zamani Esfahlani et al and Faskowitz et al. Additionally, except for a single analysis in Figure 5, Luppi et al (2022) focus exclusively on human imaging data.

Finally, we note that since our original submission, we have published a review article that more clearly outlines what we mean by “edge-centric” analyses (Betzel et al 2023; TICS). We also reference this in the manuscript.

We apologize for any confusion and have updated the manuscript to more clearly indicate what we mean by “edge-centric framework.”

The revised **Introduction** text reads:

“We refer to studies that calculate and analyze edge time series or the correlation structure of edge time series (so-called edge functional connectivity [20,21,22,23]) as ‘edge-centric’.”

- What is meant by ‘clusters’: brain areas or temporal clusters? What is a ‘centroid’, which method was used to determine ‘axonal’ projections?

We appreciate the opportunity to clarify. We used the term “cluster” generically to refer to any grouping of data where the group labels were determined using a data-driven approach. For example, we calculated the pairwise similarity between all even co-fluctuation patterns and clustered the resulting similarity matrix using a hierarchical and recursive modularity maximization (see Figure 3). In this case, the data being clustered are co-fluctuation patterns at different points in time.

We also applied the same algorithm to FC to detect functional modules/clusters (see Figure S4). In this case, the data being clustered are brain regions/parcels.

We have updated the text so that when we used words like “cluster”, “module”, “group”, or “community”, we are explicit about what data are being clustered.

To this end, we have made the following edits to the text.

In Figure 1:

“(g) Although no two events are identical in terms of co-fluctuation patterns, events can be grouped (broadly) into clusters--i.e. groups of co-fluctuation patterns whose mutual similarity to one another exceeds what would be expected by chance.”

In Figure 3:

“(b) Hierarchical clustering of co-fluctuations patterns. Gray circles represent clusters--groups of co-fluctuation patterns derived using a variant of modularity maximization--and lines indicate parent-child relationships.”

We are also happy to define the term “centroid”. In this case, centroids refer to the center-of-mass over all data points assigned to a cluster. For instance, suppose 100 co-fluctuation patterns were grouped into a single cluster. The centroid for that cluster is simply the mean co-fluctuation pattern. We now define this term when it is first used in the **Results** section.

To this end, we have updated the caption for **Figure 3**. It now reads:

“Panels d, e, and f represent centroids for the three large clusters detected at hierarchical level 2. Here, centroids refer to the mean across all co-fluctuation patterns assigned to a given cluster.”

As well as the sub-section entitled “High-amplitude events can be sub-divided into recurring network states”:

“To characterize each cluster further, we estimated and analyzed their respective centroids--i.e. the mean co-fluctuation pattern across all patterns assigned to that cluster.”

The reviewer also asks about the methods used to determine axonal projections. Briefly, all mouse structural connectivity came from a public resource made available by the Allen Brain Institute. This resource summarizes the outcomes of 428 tract-tracing experiments, in which viral tracers were injected into select brain regions in a single hemisphere of the mouse brain. The tracers labeled axons so that they expressed a green-fluorescing protein. Each brain could be serially sectioned and subsequently imaged using two-photon tomography to track pathways illuminated by the tracer.

Note that unlike connectomes reconstructed non-invasively using diffusion MRI and tractography, the procedure for reconstructing axonal projections—axons originating near the injection site and terminating elsewhere in the brain—is invasive and requires that the animal be sacrificed.

Note further that while this “tract tracing” approach is not without limitations, viral tracing of projections is nonetheless considered the “gold standard” for meso-scale connectome reconstruction. We point the reviewer to the description of the Allen Institute’s mouse connectome in the **Materials and Methods** section. For completeness, we have copied those details here:

“The mouse anatomical connectivity data used in this work were derived from a voxel-scale model of the mouse connectome made available by the Allen Brain Institute [100,101](<https://data.mendeley.com/datasets/dxtzpvv83k/2>). Here, we preserved the directionality of connections--i.e. no symmetrization step was included in the pre-/post-processing pipelines.

Briefly, the mouse structural connectome was obtained from imaging enhanced green fluorescent protein (eGFP)–labeled axonal projections derived 428 viral microinjection experiments, and registered to a common coordinate space [43]. Under the assumption that structural connectivity varies smoothly across major brain divisions, the connectivity at each voxel was modeled as a radial basis kernel-weighted average of the projection patterns of nearby injections [100]. Following the procedure outlined in [99], we re-parcellated the voxel scale model in the same 182 nodes used for the resting state fMRI data, and we adopted normalized connection density (NCD) for defining connectome edges, as this normalization has been shown to be less affected by regional volume than other absolute and/or relative measure of interregional connectivity [101].”

- What is meant by fine-scale investigations? fMRI in rodents?

We apologize for the confusion. The term “fine-scale” refers to the temporal resolution of our analyses. Edge time series, unlike other methods for detecting time-varying changes in connections’ weights, requires no windowing or smoothing kernel. Rather, the edge fluctuations are defined at a timescale equivalent to the framerate at which the data were captured – the “fine scale”.

We have updated the manuscript to clarify this terminology. The revised text reads:

*“Collectively, our findings set the stage for future, more targeted and hypothesis-driven investigations into the anatomical underpinnings of network-level co-fluctuations **at the fine-scale--i.e. a temporal resolution equivalent to that of the acquisition frame rate.**”*

Results and Methods:

- what type of anaesthesia was used? It seems like an important detail to mention for the cross-species comparison.

We regret this omission. Mice fMRI timeseries were acquired using low level halothane anesthesia (0.8%). We have updated the manuscript to reflect this detail.

The revised text in the **Materials and Methods** section reads:

“Single-shot BOLD echo planar imaging time series were acquired using an echo planar imaging sequence with the following parameters: repetition time/echo time, 1200/15 ms; flip angle, 30 degrees; matrix, 100 × 100; field of view, 2 × 2 cm²; 18 coronal slices; slice thickness, 0.50 mm; 1500 (n = 19) volumes; and a total rsfMRI acquisition time of 30 min. Mice were anaesthetized with isoflurane (5% induction), intubated and artificially ventilated (2%, surgery). The left femoral artery was cannulated for continuous blood pressure monitoring and terminal arterial blood sampling. At the end of surgery, isoflurane was discontinued and substituted with halothane (0.8%).”

- Some methodological considerations are repeated or introduced in depth in the results (for example the estimation of bipartition), whilst other critical information is omitted, for example how was static FC computed?

We appreciate this point and apologize for any confusion. We felt that many of the analyses we described would be relatively unfamiliar to the casual network neuroscience reader (edge-centric analysis, though becoming more common, is still new). Consequently, we opted to provide extra details in our descriptions of these analyses in the **Results** section. We note that their complete description is included **Materials and Methods** section.

We also apologize if we omitted critical information. For the specific case of FC definition, we follow the *de facto* standard in human fMRI imaging and define FC between regions *i* and *j* and the product-moment correlation of their activity time series. Although this was previously noted in both the **Materials and Methods** as well as the **Results** section, we now indicate this more clearly.

The revised text reads:

“One of the first observations made using edge time series was that, with only a small subset of high-amplitude frames--putative “events”--it was possible to accurately reconstruct static FC [13]. Note that here we define whole-brain FC as the matrix of all interregional correlations.) To date, these types of analyses have been carried out largely

using functional imaging data acquired from awake humans; whether such events exist in data acquired from animals has been underexplored. Here, we assess whether a similar effect is evident when we apply edge time series to functional imaging data acquired from anesthetized mice.”

- Figure 1 seems to be missing some information: E: should the label be Edge-pairs rather than edges? F: what are the different boxes? Different time points? What figure represents the RMS timeseries?

We appreciate the opportunity to clarify. We note also that Reviewer 1 raised concerns about the interpretation of Figure 1. In light of these two independent reviews highlighting the same point, we have updated the figure substantially so that it better communicates our general methodology.

We include a copy of the updated figure below, for reference, and address the reviewer’s specific concerns afterwards.

Figure 1. Schematic illustrating edge time series construction and clustering. (a) Array of parcel time series. Rows and columns correspond to parcels (ordered by anatomical division) and frames, respectively. (b) Whole-brain functional connectivity is typically estimated as the correlation matrix of parcel time series. That is, the weight of the functional connection between nodes i and j is specified as the product-moment correlation coefficient, r_{ij} . (c) The procedure for estimating r_{ij} entails z-scoring each parcel time series, calculating their elementwise product—i.e. $z_i(1) \times z_j(1), \dots, z_i(T) \times z_j(T)$ —and taking the mean of those products (sum divided by the number of samples). Omitting the averaging step yields the co-fluctuation (or “edge”) time series $r_{ij}(t) = [z_i(1) \times z_j(1), \dots, z_i(T) \times z_j(T)]$, whose elements encode time-varying changes in the weight of the connection between nodes i and j . In this panel, we show time series for regions i and j (green and blue curves) and their corresponding edge time series (gray). (d) We can calculate edge time series for all node pairs (edges) in the network and arrange them as rows in an edge-by-frame matrix. (e) Previous studies identified infrequent and short-lived high-amplitude bursts. These bursts or “events” can be detected by calculating the root mean square (RMS) across all edges at each frame and identifying peaks whose amplitude exceeds that of a null distribution (estimated using same procedures as empirical RMS, but starting with circularly shifted parcel time series). (f) The co-fluctuation patterns expressed during events are very different than those expressed during low-amplitude frames. Here, we highlight approximately 175 frames and show whole-brain co-fluctuation matrices for three local maxima (events; red border) and three local minima (troughs; gray border). (g) Although no two events are identical in terms of co-fluctuation patterns, events can be grouped (broadly) into clusters. We detect them by computing the similarity (concordance) between all pairs of events and directly clustering the resulting matrix using a hierarchical algorithm. Here, we highlight two large clusters and their respective centroids (the mean co-fluctuation pattern across all events assigned to each cluster).

In what was previously labeled Figure 1E (now Figure 1E), we show edge time series. Each row therefore corresponds to a node pair (a single edge), so the current label is accurate.

Regarding Figure 1f. The matrices depicted there are co-fluctuation matrices corresponding to peaks and troughs in the RMS signal. We now include a “blown up” plot of the RMS signal, with the red peaks (events) and gray troughs highlighted.

The RMS signal was shown in the original version of Figure 1 but not clearly labeled. We now label the RMS signal in Figure 1E.

Note that we have also updated the figure and the caption to better explain what is depicted there.

- Is static FC definition part of this figure as Figure 2 is referenced only later in the text?

We show the static FC matrix in Figure 1B because of its relationship with edge time series. If we define FC to be an interregional correlation, then the mean of the edge time series for the edge $\{i, j\}$ is *exactly* the weight of the connection between those same nodes.

Note that, based on these previous comments, we also define FC in the text, as well.

The revised text, which we referenced previously in this same response letter, reads:

“One of the first observations made using edge time series was that, with only a small subset of high-amplitude frames--putative “events”--it was possible to accurately reconstruct static FC [13]. Note that here we define whole-brain FC as the matrix of all interregional correlations. To date, these types of analyses have only been carried out using functional imaging data acquired from awake humans; whether such events exist in data acquired from animals is unknown. Here, we assess whether a similar effect is evident when we apply edge time series to functional imaging data acquired from anesthetized mice.”

- Figure 2: What are the axis labels for the scatter plots?

The axis labels are connection weights. We have updated the figure with clearer labels. For completeness, we show the updated figure below:

- Figure 3: what is EV?

EV refers to eigenvalue. We now write that out rather than express it as an abbreviation. We show the updated figure below.

- Figure 4: Could the hippocampus be annotated in the graph? What does the size of each node in the dot-plot suggest?

We now include arrows pointing to the hippocampal nodes in panels b, e, and h. We have also updated the figure caption to explain what the size of each point represents.

We paste the figure along with the updated figure caption below:

Figure 4. **Linking high-amplitude events to structural connectivity.** Panels *a*, *d*, and *g* represent bipartition communities for each of the three largest event cluster centroids in hierarchical level 2. Panels *b*, *e*, and *h* force-directed layouts of the induced sub-graph containing only nodes in either of the bipartition communities. **The size of nodes are proportional to their weighted degree (strength).** Panels *c*, *f*, and *i* show the induced modularity of each sub-graph.

- Figure S2: Could the abbreviations be introduced?

We apologize for this omission. We now spell out the anatomical labels rather than expressing them as abbreviation. We include, here, the updated figure.

- Figure S7: What does the abbreviation Ind. stand for?

We apologize for the confusion. The abbreviation “Ind.” refers to a geometry independent null model, i.e. a permutation that does not preserve the spatial relationships between nodes in the network and their connectivity. This contrasts with the “geometry preserving” null model that seeks permutations of the data that approximates the spatial variogram.

We have now updated the figure legend, replacing “Ind” with “Geometry independent” and “Geom” with “Geometry preserving”.

We include, for reference a copy of the updated figure here.

- the comparison of axonal tracing and dMRI tractography in humans bears limitations that should be discussed.

We agree that it would be useful to discuss the distinctions between the two techniques, as the resulting structural connectivity matrices have different sets of features.

Most saliently, tract tracing is invasive. It requires the injection of viral tracers directly into the brain and imaging the tracer requires the animal be sacrificed. In contrast, dMRI data is

acquired non-invasively. Similarly, the reconstruction of white-matter tracts (streamlines) using tractography algorithms can be performed without sacrificing the animal/human.

Whereas tractography infers the presence of connections indirectly, the invasive nature of the measurements made using tract tracing make it the “gold standard” for reconstruction of long-distance meso-scale connections. It can also resolve directed connections; tractography can identify principal axes of diffusion but cannot distinguish directionality. Consequently, networks reconstructed from tract tracing data can be asymmetric such that $A_{ij} \neq A_{ji}$.

In addition, white-matter networks reconstructed from dMRI and tractography exhibit known biases, e.g. crossing fibers (voxels with fibers oriented at different angles) are notoriously difficult to resolve as are superficial white-matter connections. Moreover, different tractography algorithms are sensitive to parameter choices and can result in networks of very different character. For instance, probabilistic tracking typically yields denser networks than deterministic tracking (greater proportion of possible connections exist), which impacts subsequent network measures.

Despite its limitations, dMRI + tractography boasts several advantages over tract-tracing. As noted early, it can be acquired non-invasively. Most importantly, whereas constructing a whole-brain network of inter-areal projections through tract-tracing studies requires hundreds of experiments (and therefore sacrificing many animals), whole-brain white-matter networks can be reconstructed on a per-subject basis using dMRI + tractography. That is, the mouse connectome reconstructed from tract-tracing experiments yields one connectivity matrix that, by necessity, is treated as universal (the same matrix for every subject). The human connectome, on the other hand, is reconstructed for each participant is therefore specific to that person.

In short, both tract tracing and dMRI + tractography have clear advantages and disadvantages. The mouse tract-tracing data are closer to the “ground truth” than the human data, but lack subject specificity. The human data is likely “noisier” but is reconstructed at the level of individual subjects. Here, we report converging results – networks generated using both techniques exhibit similar relationships with respect to event co-fluctuation patterns. We now discuss the relative advantages/disadvantages of these approaches.

The newly added text reads:

“Our study relies on connectome data reconstructed using two distinct techniques: invasive tract-tracing in mice and non-invasive tractography and diffusion MRI in humans. Although we report converging evidence across both modalities showing that high-amplitude events are underpinned by modular structural networks, there are a number of dissimilarities worth noting explicitly. Critically, the mouse connectome is directed, i.e. $W_{ij} \neq W_{ji}$, and capable of resolving asymmetries in connection weight, whereas the human connectome is not. Additionally, it is well-established that tractography algorithms struggle to resolve “crossing fibers” (Wedeen et al 2008),

recover superficial tracts (Reveley et al 2015), and, even across well-established pipelines, can lead to variability in tract reconstructions (Maier et al 2017). Nonetheless, the non-invasive nature of MRI means that human connectomes can be reconstructed at a whole-brain level for individual participants. In summary, we identify comparable effects using both techniques, but note that in future studies, it may be advantageous to focus on dissimilarities--e.g. the specific contributions of directed connections."

REVIEWERS' COMMENTS:

Reviewer #1 (Remarks to the Author):

I thank the authors for taking my comments into consideration.
The methodological aspects now read more smoothly.
I have no further comments.
Congratulations on this work

Reviewer #2 (Remarks to the Author):

The authors have done an excellent job with my comments. I am very happy to endorse this manuscript for publication.

Reviewer #3 (Remarks to the Author):

The authors have substantially revised the manuscript to improve clarity, which was my main concern. They have also addressed and explained the other points that were raised and improved the figures. Taken together, I now recommend this paper for publication.